# Ribosome inhibition by *C9ORF72*-ALS/FTD-associated poly-PR and poly-GR proteins revealed by cryo-EM

Anna B. Loveland [1], Egor Svidritskiy[1], Denis Susorov[1], Soojin Lee[2], Alexander Park[1], Sarah Zvornicanin[1], Gabriel Demo [1,3], Fen-Biao Gao [2✉] & Andrei A. Korostelev [1✉]

Toxic dipeptide-repeat (DPR) proteins are produced from expanded $G_4C_2$ repeats in the *C9ORF72* gene, the most common genetic cause of amyotrophic lateral sclerosis (ALS) and frontotemporal dementia (FTD). Two DPR proteins, poly-PR and poly-GR, repress cellular translation but the molecular mechanism remains unknown. Here we show that poly-PR and poly-GR of ≥20 repeats inhibit the ribosome's peptidyl-transferase activity at nanomolar concentrations, comparable to specific translation inhibitors. High-resolution cryogenic electron microscopy (cryo-EM) reveals that poly-PR and poly-GR block the polypeptide tunnel of the ribosome, extending into the peptidyl-transferase center (PTC). Consistent with these findings, the macrolide erythromycin, which binds in the tunnel, competes with poly-PR and restores peptidyl-transferase activity. Our results demonstrate that strong and specific binding of poly-PR and poly-GR in the ribosomal tunnel blocks translation, revealing the structural basis of their toxicity in *C9ORF72*-ALS/FTD.

[1] RNA Therapeutics Institute, UMass Chan Medical School, 368 Plantation Street, Worcester, MA 01605, USA. [2] Department of Neurology, UMass Chan Medical School, 368 Plantation Street, Worcester, MA 01605, USA. [3] Central European Institute of Technology, Masaryk University, Kamenice 5, Brno 625 00, Czech Republic. ✉email: Fen-Biao.Gao@umassmed.edu; Andrei.Korostelev@umassmed.edu

Frontotemporal dementia (FTD), the second most common form of dementia, is caused by focal degeneration of the prefrontal and/or temporal lobe, resulting in changes in personality, social behaviors, language production, and other deficits[1]. FTD shares extensive pathological, genetic, and molecular similarities with the motor neuron disease amyotrophic lateral sclerosis (ALS)[2–4]. The most prevalent cause of both diseases is the expansion of GGGGCC ($G_4C_2$) repeats in the first intron of the *C9ORF72* gene[5,6]. Unaffected individuals typically carry fewer than twenty $G_4C_2$ repeats, whereas *C9ORF72*-ALS/FTD patients may have up to hundreds or thousands of $G_4C_2$ repeats[7,8].

It remains unclear how the expanded $G_4C_2$ repeats cause the diseases. Wildtype C9ORF72 protein is produced despite the expanded repeats, albeit at a lower level due to hypermethylation of one of the promoters[5,9,10]. Complete knockout of *C9orf72* in mice does not cause neurodegeneration, arguing that deficiency in C9ORF72 protein expression or function is unlikely to be fully responsible for ALS/FTD pathogenesis[11–13]. By contrast, the products of transcription and translation of sense and antisense $G_4C_2$ repeat sequences have been observed in patient cells and proposed to be pathogenic[14–16]. Although the molecular mechanisms by which $G_4C_2$ repeats-containing RNAs are produced and translated remain largely unclear[17], these RNAs are found in the cytosol and associated with polysomes[18,19]. Of the five resulting dipeptide-repeat (DPR) proteins, poly-proline-arginine (poly-PR) and poly-glycine-arginine (poly-GR) are most highly toxic when ectopically expressed in cellular and animal models[20–27]. Distribution of poly-GR correlates with neurodegeneration in human patient brains[28–30]. Poly-PR is thought to be so toxic that many human patient neurons expressing this DPR protein may be lost by the time of autopsy[22]. Although these and other DPR proteins were proposed to interfere with numerous cellular pathways[3,7,20,31–35], a specific molecular and structural mechanism of action of the arginine-rich DPR proteins remains unknown.

Biochemical and interactome analyses revealed that poly-PR and poly-GR bind to the translational machinery[24,33,36–43] and impair global cellular translation[25,36,39,41,44,45]. Poly-PR and poly-GR colocalize with ribosomes in patient brain tissues[39] and repress translation in several cellular and mouse models[25,36,41,44,45]. These DPR proteins accumulate in the cytoplasm and cellular organelles[20,22,24,39,40], and impair global translation when added to cells and cell extracts in trans[20,41,45]. In addition, translational stalling occurs *in cis* on the $G_4C_2$ repeat mRNA encoding poly-PR and poly-GR repeats, which is regulated by ribosome quality control mechanisms[40,46]. While translation inhibition by poly-PR and poly-GR might be a predominant pathway underlying cellular toxicity, the molecular and structural mechanisms of this phenomenon are not known.

In this work, we used biochemical analyses and cryogenic electron microscopy (cryo-EM) to identify the specific ribosomal binding site and structural mechanism for translation inhibition by poly-PR and poly-GR, accounting for cellular toxicity of these DPR proteins in *C9ORF72*-ALS/FTD.

## Results

**Poly-PR and poly-GR inhibit translation.** To understand how poly-PR and poly-GR disrupt translation in trans, we first used mammalian cell lysates (rabbit reticulocyte lysates, RRL) to translate nanoluciferase or firefly luciferase and followed their light emission in the presence of DPR proteins over time. Both $PR_{20}$ and $GR_{20}$ (*the subscript denotes the number of PR or GR repeats*) strongly repress translation (Fig. 1a, top 2 panels, Supplementary Fig. 1a, b) with half-maximal inhibition at ~0.7 μM, consistent

with previous studies showing inhibition of translation in cellular lysates by these DPR proteins[41,43,45]. By contrast, 10 μM $GP_{20}$ (glycine-proline)—another soluble translation product of the $G_4C_2$ expansion—does not inhibit translation (Supplementary Fig. 1c). Translation is also not inhibited by 100 μM L-arginine, indicating that the oligopeptide context of positively-charged arginine residues is critical for translation inhibition (Supplementary Fig. 1d). Moreover, poly-PR and poly-GR do not interfere with luciferase enzyme activity or aggregate luciferase mRNAs at these inhibitory concentrations (Supplementary Fig. 1e–g). Thus, poly-PR and poly-GR specifically inhibit translation in a mammalian cell lysate.

Translation could be inhibited at several steps, including initiation, elongation, and termination/recycling[47]. The nanoluciferase assay in this work allows to monitor all steps, because nanoluciferase luminescence requires the complete translation of the nanoluciferase mRNA, including nanoluciferase release from the ribosome at the stop codon[48]. To elucidate the steps affected by the DPR proteins, we compared translation inhibition by $PR_{20}$ and $GR_{20}$ (Fig. 1a, b, panels 1–2. Supplementary Fig. 1m–o) to that by eukaryotic translation inhibitors with established modes of action (Fig. 1a, b, panels 3–5). Harringtonine (homoharringtonine) primarily stalls the initiating 80S ribosomes by binding the A-site of the 60S subunit and interfering with formation of the first peptide bond[49,50]. Cycloheximide inhibits translation elongation by binding the E-site of the 60S subunit and thus preventing translocation of deacylated tRNA from the P to E-site[47,50]. Finally, the non-hydrolyzable GTP analog, GDPCP, inhibits all steps of translation by binding the GTPases that catalyze each step[51]: initiation (eIF2 and eIF5B), elongation (eEF1A and eEF2), and termination (eRF3). As expected, these inhibitors result in different profiles of nanoluciferase activity. Preincubating RRL with harringtonine prior to adding mRNA (Fig. 1a, panel 3) strongly inhibits translation, but does not alter the time at which the initial nanoluciferase signal appears, in keeping with the lack of elongation inhibition. Indeed, when added after the appearance of nanoluciferase signal, i.e. after translation initiation, harringtonine fails to affect translation (Fig. 1b, panel 3). By contrast, cycloheximide delays the appearance of nanoluciferase luminescence and reduces the rate of translation, consistent with inhibition of elongation (Fig. 1a, b, panel 4). Inhibition by GDPCP follows a pattern similar to that of cycloheximide (Fig. 1a, b, panel 5), although the maximal luciferase signal is substantially reduced at higher concentrations of GDPCP, in keeping with inhibition of all translation steps including protein release. The inhibition profiles of $PR_{20}$ (Fig. 1a, b, panel 1) and $GR_{20}$ (Fig. 1a, b, panel 4) resemble those of cycloheximide and GDPCP. Addition of these inhibitors ($PR_{20}$, $GR_{20}$, cycloheximide, and GDPCP) after translation initiation (Fig. 1b) inhibits nanoluciferase production within seconds, indicating that $PR_{20}$ and $GR_{20}$ can inhibit translation elongation.

To further investigate the inhibition modes of $PR_{20}$ and $GR_{20}$, we performed polysome profiling in RRL containing endogenous globin mRNAs in the presence or absence of the five inhibitors (Fig. 1c). This approach distinguishes the stalling of the initiating 80S ribosome, such as that induced by harringtonine (Fig. 1c, panel 3), from the stalling of elongating polysomes by elongation inhibitors, such as cycloheximide (Fig. 1c, panel 4). Notably, GDPCP not only stabilizes polysomes but also shows an increase in half-mer peaks (Fig. 1c, panel 5, see black arrows), which appear as addition of "half a ribosome" between ribosome peaks, indicating accumulation of pre-initiation ribosomal 40S subunits halted prior to subunit joining[51,52]. $PR_{20}$ and $GR_{20}$ result in the appearance of half-mer peaks in the polysome fraction, resembling those formed by GDPCP, and echoing the observation of half-mers in HeLa cells treated with $PR_{20}$[43]. The shapes of the

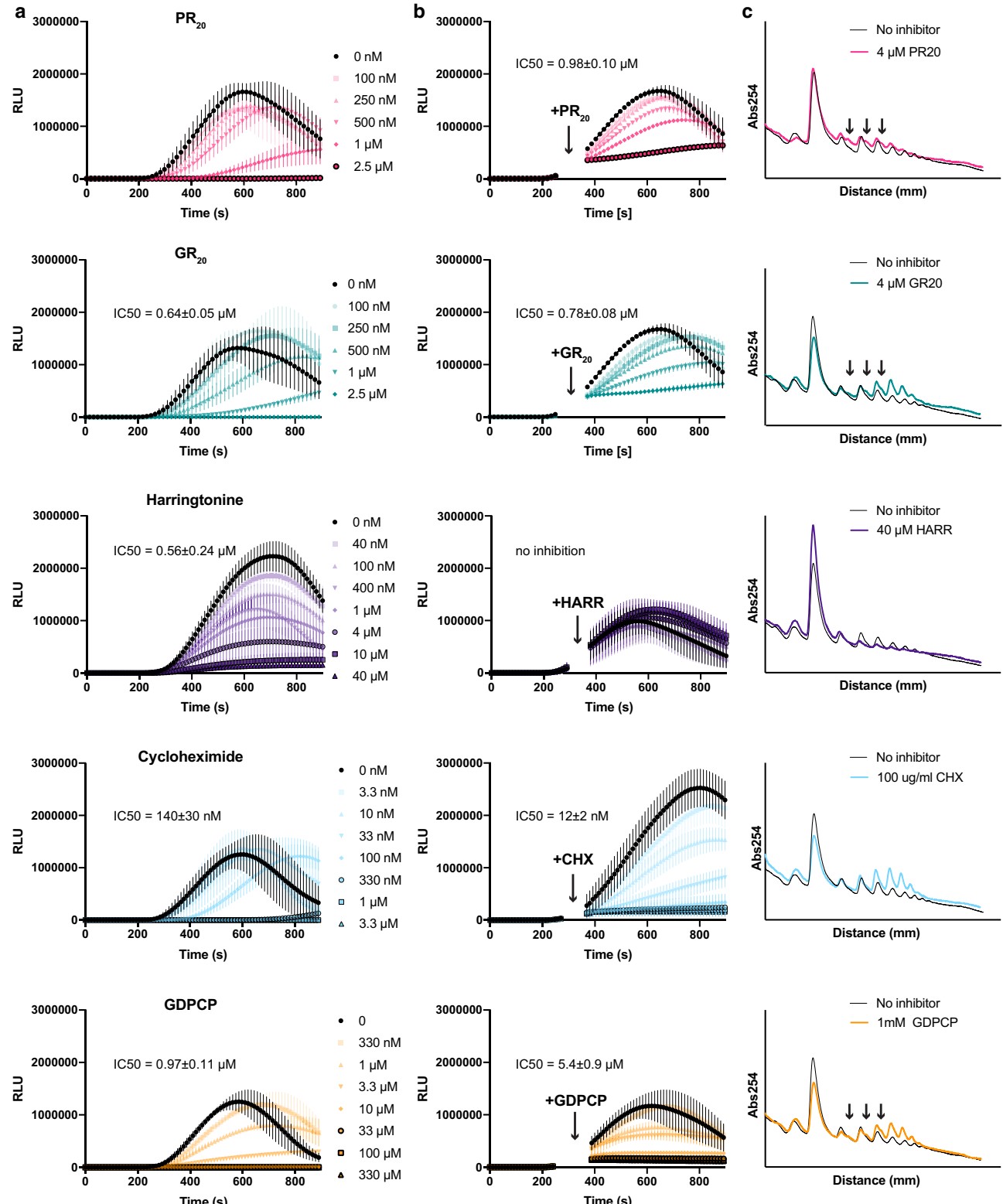

polysome profiles formed with PR$_{20}$ (Fig. 1c, panel 1) and GR$_{20}$ (Fig. 1c, panel 2) differ by the heights of the 80S and polysome peaks, suggesting that the modes of translation inhibition by these two DPR proteins may differ (see also Supplementary Fig. 1h–l). Indeed, toe-printing of polysome fractions shows that ribosomes accumulate at the start codon in the presence of PR$_{20}$ and harringtonine, but ribosomes accumulate within the open reading frame in the presence of GR$_{20}$, cycloheximide, and GDPCP

(Fig. 2). These data indicate that PR$_{20}$ functions as both a translation initiation and elongation inhibitor, whereas GR$_{20}$ predominantly inhibits elongation.

**Poly-PR and poly-GR inhibit ribosomal peptidyl transferase.** We reasoned that PR$_{20}$ and GR$_{20}$ could strongly inhibit translation by disrupting the central functional site of the ribosome

**Fig. 1 Poly-PR and Poly-GR inhibit mammalian translation, comparable to some small-molecule translation inhibitors. a** Comparison of translation inhibition by $PR_{20}$ and $GR_{20}$ to that by harringtonine, cycloheximide, and GDPCP upon a 5-min preincubation with rabbit reticulocyte lysates prior to addition of nanoluciferase mRNA ($t = 0$). IC50 of $PR_{20}$ and $GR_{20}$ are similar to those of cycloheximide, harringtonine, and GDPCP ($n = 3$ independent experiments, mean ± SEM; RLU, relative luminescence units). **b** Comparison of translation inhibition by $PR_{20}$ and $GR_{20}$ to that by harringtonine, cycloheximide, and GDPCP. The inhibitors were added 300–400 s (gaps in the curves) after addition of nanoluciferase mRNA. $PR_{20}$ and $GR_{20}$ act similarly to elongation inhibitors cycloheximide and GDPCP ($n = 3$ independent experiments, mean ± SEM). **c** Comparison of polysome profiles of rabbit reticulocyte lysate translating endogenous mRNA in the presence of translation inhibitors (Abs254, UV absorbance at 254 nm). The translation mixture was incubated for 5 min at 30 °C in the presence or absence of $PR_{20}$, $GR_{20}$, harringtonine, cycloheximide, and GDPCP ($n = 2$ independent experiments, representative traces with concurrent uninhibited control are shown). Half-mer peaks (black arrows) are most prominent with $PR_{20}$, $GR_{20}$, and GDPCP. Source data are provided as a Source Data file.

conserved in all organisms: the peptidyl-transferase center (PTC). To test if translational repression is due to the direct inhibition of peptide bond formation, we first used bacterial 70S ribosomes, a robust system for in vitro kinetic studies[53,54], and then adapted the kinetic assay for mammalian 80S ribosomes. To this end, we measured *E. coli* ribosome-catalyzed peptidyl transfer from peptidyl-tRNA to puromycin (an aminoacyl-tRNA mimic; Methods). While L-arginine does not affect the peptidyl transfer reaction (Supplementary Fig. 2b), $PR_{20}$ and $GR_{20}$ strongly inhibit the reaction with apparent inhibition constants ($K_i$) of $44 \pm 9$ nM and $164 \pm 31$ nM, respectively (Fig. 3a, b). The longer $PR_{40}$ confers stronger inhibition ($K_i \leq 30$ nM). By contrast, shorter peptides $GR_{10}$ and $PR_{10}$ exhibit a weaker inhibitory effect ($K_i$ of $0.6 \pm 0.3$ μM and $1.0 \pm 0.3$ μM, respectively; Fig. 3b) and high concentrations of shorter peptides $PR_4$ and $GR_4$ (2 μM) fail to inhibit peptidyl transfer (Supplementary Fig. 2d, e). These findings demonstrate that poly-PR and poly-GR inhibit the peptidyl-transferase activity of bacterial ribosomes. We also tested whether these DPR proteins inhibit translation in bacterial cell lysates. Unlike RRL, translation in *E. coli* lysate was resistant to $PR_{20}$ below 5 μM and to $GR_{20}$ up until 10 μM $GR_{20}$ (Supplementary Fig. 2a), suggesting that the bacterial lysate contains components that interact with the DPR proteins and mitigate their inhibitory effect on *E. coli* ribosomes.

We performed the puromycin assay using purified rabbit ribosomes, whose functional and structural properties are nearly identical to those of human ribosomes. We found that $PR_{20}$ starts to inhibit peptidyl transfer at ~100 nM, comparable to or exceeding that of PTC-binding small-molecule inhibitors of mammalian translation, such as harringtonine[55,56] and nagilactone C[57] (Fig. 3c). $GR_{20}$ is also inhibitory, albeit to a lower extent than $PR_{20}$ (Supplementary Fig. 2f). The longer $PR_{40}$ abolishes peptidyl transfer ($K_i \leq 30$ nM), whereas the shorter $PR_{10}$ confers no inhibition at 2 μM (Supplementary Fig. 2g, h). Thus, the efficiency of inhibition increases with the length of DPR proteins, consistent with a disease-causing length threshold[7,8]. Our observation of inhibited peptidyl transfer on both bacterial 70S and mammalian 80S ribosomes suggests that longer poly-PR and poly-GR bind a conserved, functionally critical region of the translation machinery. Sub-micromolar IC50 and $K_i$-values suggest that the binding is strong and that the binding results in the direct (e.g., competition with ribosomal substrates) or indirect inhibition of the PTC activity.

**Poly-PR and poly-GR bind the polypeptide tunnel of eukaryotic ribosomes**. To understand where and how the DPR proteins bind to the ribosome, we obtained near-atomic resolution cryo-EM structures of $PR_{20}$ and $GR_{20}$ bound to eukaryotic 80S ribosomes (yeast and rabbit) and $PR_{20}$ bound to the bacterial 70S ribosomes (Figs. 4, 5, Supplementary Figs. 3–5). The most resolved 2.4-Å structure of the yeast 80S•tRNA•$PR_{20}$ complex reveals poly-PR bound to the polypeptide tunnel and stretching into the PTC, thus excluding tRNA (Fig. 4a–c, Supplementary

Fig. 3a, c). The poly-PR chain traverses the tunnel with its N-terminus directed toward the 60S tunnel exit. Nine dipeptide repeats of poly-PR are resolved from the PTC through the polypeptide tunnel constriction at the universally conserved 25S rRNA residue A2404 (A3908 in *H. sapiens*; A2062 in *E. coli*) (Fig. 4b) toward a second constriction between A883 (A1600 in *H. sapiens*; A751 in *E. coli*) and protein uL4 (Fig. 4c), and then toward eL39 and uL22. Arginine and proline residues of $PR_{20}$ stack against 25S rRNA nucleotides, or aromatic or arginine residues of uL4 (Fig. 4b, Supplementary Fig. 3h), in keeping with strong binding of poly-PR. Several arginine residues are also stabilized by negatively charged phosphates of 25S rRNA.

In the PTC, poly-PR would clash with an aminoacyl or peptidyl moiety on P-site tRNA. Lower-resolution features continue away from the PTC toward uL16 and into the intersubunit space typically occupied by A-site tRNA during translation (Supplementary Fig. 3f). In the opposite direction—toward the 60S subunit solvent-exposed surface—the widening polypeptide tunnel also shows lower-resolution features corresponding to a continuous poly-PR chain (Supplementary Fig. 3f). Together, the ordered and less ordered regions account for ~15 PR repeats, rationalizing how the longer poly-PR chains threads the tunnel and inhibits translation by preventing A- and P-tRNA binding in the PTC.

The structure of the polypeptide tunnel, where poly-PR binds, is nearly fully conserved between yeast and mammals (Supplementary Fig. 3h). Indeed, our 3.1-Å resolution cryo-EM structure of the rabbit 80S•tRNA•$PR_{20}$ complex, confirms that poly-PR binds similarly in the mammalian and yeast ribosomes (Fig. 4b, d, Supplementary Fig. 4). Poly-PR density is strongest in the tunnel constriction near A3908 of 28S rRNA. From here, poly-PR reaches into the PTC and either competes with P-site tRNA (Fig. 4f, Supplementary Fig. 4g, i) or packs on the ribose of the 3′ terminal nucleotide of deacylated tRNA (Supplementary Fig. 4h). Poly-PR is incompatible with aminoacylated or peptidyl P-tRNA (Supplementary Fig. 4j), suggesting that the DPR protein would compete with both the initiating and elongating tRNA substrates.

Like poly-PR, $GR_{20}$ binds in the polypeptide tunnel of both the mammalian (Fig. 4e, Supplementary Fig. 4k–q) and yeast ribosomes (Supplementary Fig. 3b, c, e). In the 2.9-Å resolution cryo-EM structure of the rabbit 80S•tRNA•$GR_{20}$ complex, poly-GR density is most continuous between the PTC and the first tunnel constriction (Fig. 4e, Supplementary Fig. 4l–p). In keeping with high flexibility of glycine residues, however, poly-GR density is less well defined than that of the rigid poly-PR and prevents unambiguous modeling of $GR_{20}$. The $GR_{20}$ density in the PTC is incompatible with the methionyl moiety of the initiating tRNA and with longer peptidyl-tRNA. Furthermore, poly-GR destabilizes the CCA end of deacyl-tRNA in the P-site, in comparison to the well-resolved CCA end in the cryo-EM data without the DPR proteins (negative control, Supplementary Fig. 4). Thus, cryo-EM data provide the structural basis for competitive inhibition of peptidyl transfer by both poly-PR and poly-GR (Fig. 4), consistent with biochemical results (Figs. 1–3).

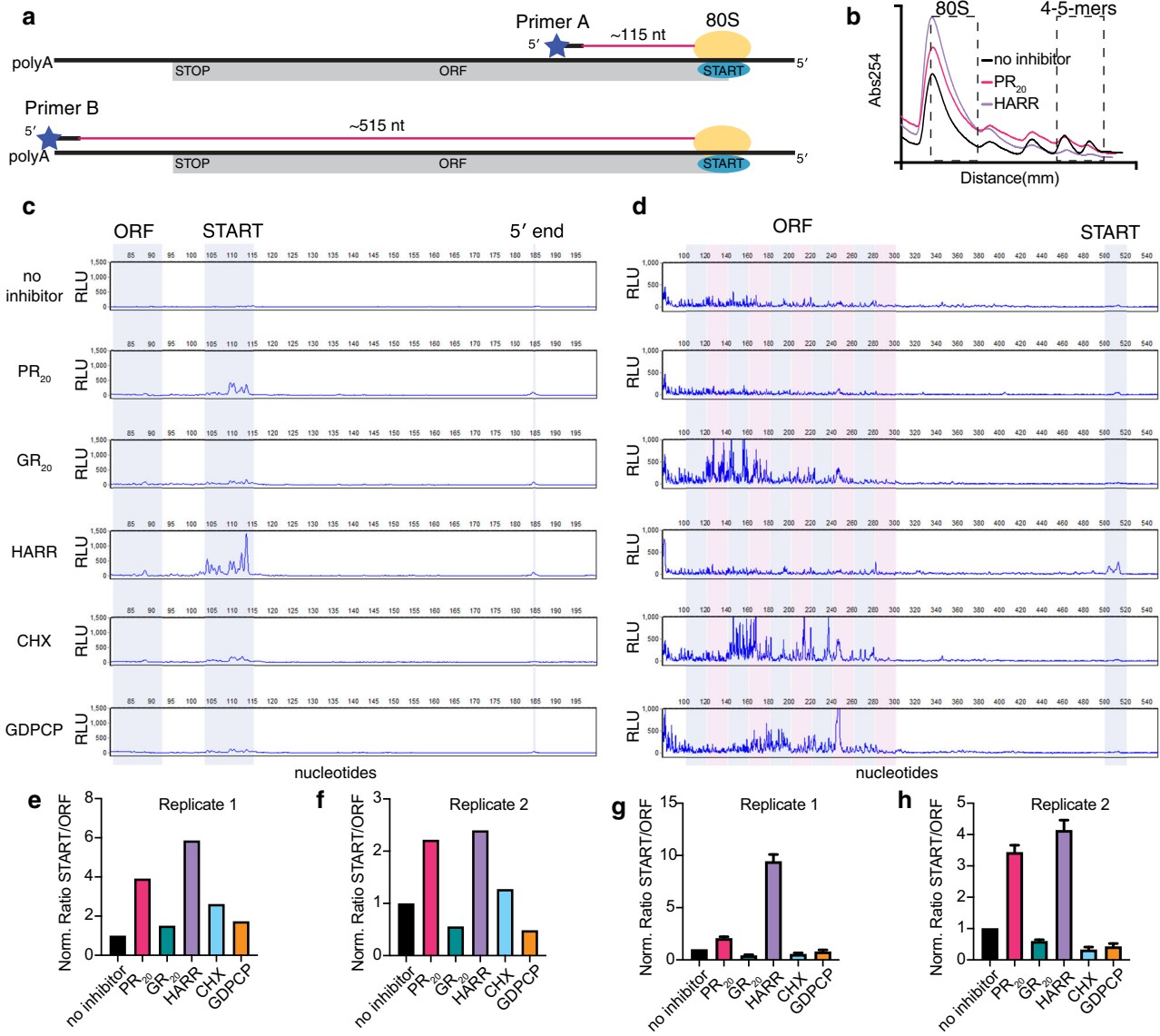

**Fig. 2 Toeprinting analyses identify ribosome stalling sites with $PR_{20}$ and $GR_{20}$. a** Scheme of toeprinting experiments using two fluorescently-labeled primers, primer A and primer B. Primer extension through the open reading frame (ORF) to ribosomes stalled at the start codon is expected to yield a peak at ~115 nt for Primer A or at ~515 nt for Primer B. **b** Following a 5-min translation reaction at 37 °C in the absence or presence of inhibitors (no inhibitor, 4 µM $PR_{20}$, 4 µM $GR_{20}$, 40 µM harringtonine, 100 µg/ml cycloheximide or 100 µM GDPCP), sucrose-gradient fractionation was used to separate free mRNA from various ribosome fractions. The 80S fraction and 4–5-mer fractions (dashed boxes) were used in assays shown in panels **b**, **c**. The sucrose gradient traces are shown for no inhibitor, $PR_{20}$ and harringtonine. **c** The 80S fraction was subjected to toeprinting analysis with primer A via fragment analyzer (see Methods). Traces reveal peaks at the expected location of the start codon. The blue-shaded zones of the traces were quantified for panel **e**, indicating enrichment of ribosomes stalled at the start codon relative to a region of the ORF especially in samples treated with harringtonine and $PR_{20}$. **d** As in (**c**) except that the 4-5mer polysome fraction was used for toe-printing analysis with primer B. Traces show many peaks in the ORF of the mRNA and also at the start codon. Blue and pink boxes indicate sections of the trace used in quantification in panel **g**. **e** The normalized ratio of start-codon-associated toeprints to coding-region-associated toeprints in the 80S fraction using primer A. **f** Independent repeat of toeprinting reaction as in **e**. **g** The normalized ratio of start-codon-associated toeprints to coding-region-associated toeprints. Ten different windows (20 bp in length each) within the coding region were quantified relative to a 20 bp window around the start codon, and the mean ± SEM is shown. **h** Independent repeat of toeprint reaction as in **g**. Source data are provided as a Source Data file.

Maximum-likelihood classification of our cryo-EM datasets revealed that all yeast and mammalian 80S ribosome states contained poly-PR or poly-GR in the polypeptide tunnel and the PTC (Supplementary Figs. 3, 4), in contrast to the negative-control 80S complex assembled without the DPR proteins (Supplementary Fig. 4a–d). These observations are consistent with the high affinity of these DPR proteins for the ribosome, implied by the IC50s and puromycin kinetics measured in our biochemical experiments. An additional density is present in the

rabbit 80S•tRNA•$GR_{20}$ map, which is not present in the control 80S•tRNA and other complexes analyzed in this work. The density at the intersubunit bridge B6[58] near protein eL24 might account for four GR repeats in a subset of 80S particles (Supplementary Fig. 4q). Alternatively, this density may be part of the eL24 C-terminal domain (residues 64 onward), for which no unambiguous density could be found in our maps. Either direct binding of $GR_{20}$ at this location or $GR_{20}$-mediated eL24 conformational change could interfere with intersubunit rotation

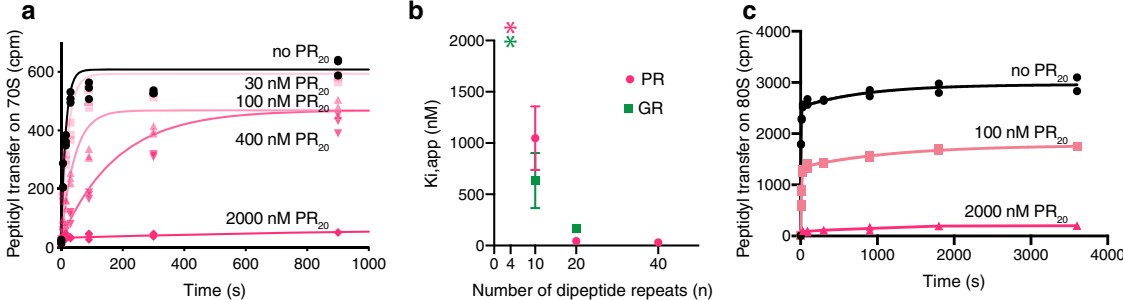

**Fig. 3 Inhibition of peptidyl transfer by PR$_{20}$ and GR$_{20}$. a** Time progress curves of inhibition of puromycin peptide bond formation by different concentrations of PR$_{20}$ on *E. coli* 70S ribosomes ($n = 3$ independent experiments). Count per minute (cpm). **b** Dependence of inhibition of peptide bond formation on the lengths of poly-PR and poly-GR repeats (puromycin assay on *E. coli* ribosomes; apparent inhibition constants are shown on the *y* axis; $n = 2$ independent experiments, error bars are standard error). $K_{i,app}$ was >2000 nM (the highest concentration tested) for both PR$_4$ and GR$_4$ and is shown as an asterisk. **c** Time progress curves of inhibition of puromycin peptide bond formation by PR$_{20}$ on rabbit 80S ribosomes ($n = 2$ independent experiments). Source data are provided as a Source Data file.

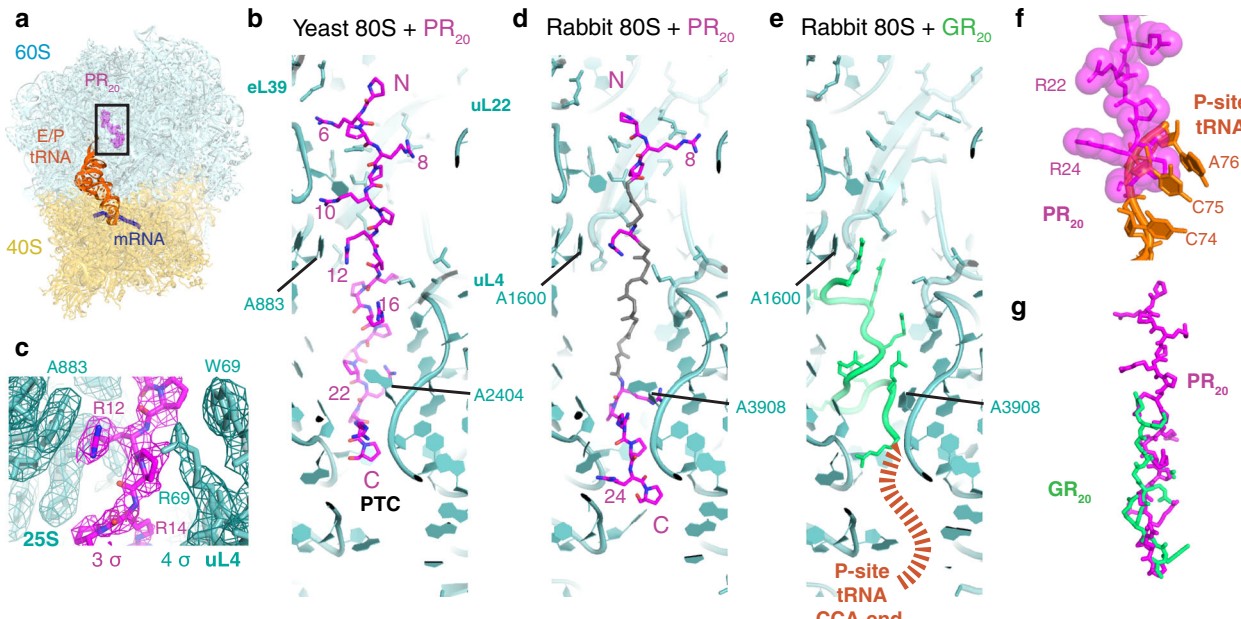

**Fig. 4 PR$_{20}$ and GR$_{20}$ bind the polypeptide tunnels of eukaryotic 80S ribosomes. a** Overview of the cryo-EM structure of yeast 80S•tRNA•PR$_{20}$ complex. **b** Cryo-EM structure of poly-PR (magenta) in the 80S tunnel (cyan); see also Supplementary Fig. 3 showing the fit of the model to the cryo-EM densities. Peptidyl-transferase center (PTC), ribosomal proteins and nucleotides, and PR repeats are labeled. **c** Poly-PR and interactions at the tunnel constriction at 25S rRNA nucleotide A883. Cryo-EM map (mesh) was B-factor sharpened (-32 Å$^2$), filtered to 2.4 Å resolution, and is shown at 3 σ for PR$_{20}$ and 4 σ for the ribosome. **d** In the rabbit 80S ribosome, PR repeats (magenta) are well resolved at the constrictions near the PTC and uL22 and are connected by lower-resolution density (modeled as gray backbone; see also Supplementary Fig. 4f–i showing the fit of the model to the cryo-EM densities). **e** In the rabbit 80S ribosome, GR repeats (green) bind near the PTC and stretch toward uL4 and destabilize binding of the CCA end of P-tRNA. **f** Steric clash between poly-PR and superimposed peptidyl-tRNA ((orange tRNA 3' end, from PDB:5LZS), superimposed via 28S rRNA[99]) on the rabbit 80S ribosome suggests poly-PR competes with P-tRNA for binding to the PTC. **g** Poly-GR occupies the same region of the peptide tunnel as poly-PR (rabbit GR vs. yeast PR structures are shown).

required during all three translation steps (initiation, elongation, and termination) and may account for an additional mode of translation inhibition by poly-GR. We cannot exclude binding of poly-PR or poly-GR to other regions, such as less well-resolved peripheral regions of the ribosome. However, neither the putative GR site at the intersubunit interface nor the potential distant interaction sites are likely to strongly interfere with peptidyl transfer. Importantly, the classifications also revealed that these two DPR proteins bind the polypeptide tunnels of isolated yeast and mammalian 60S large ribosomal subunits (Supplementary Fig. 3d, e, Supplementary Fig. 4i, n). Thus, poly-PR and poly-GR binding to the tunnel does not depend on the small 40S subunit and may affect the steps before peptidyl transfer, such as

ribosome biogenesis and/or subunit association during translation initiation, consistent with the observation of half-mers in the polysome profiles (Fig. 1c).

**Cryo-EM and competition assay of PR$_{20}$ on 70S ribosomes corroborate direct PTC inhibition.** To test whether PR$_{20}$ similarly binds the conserved PTC in bacterial ribosomes to inhibit peptidyl transfer (Fig. 3a), we performed cryo-EM analyses of a complete GTPase-catalyzed elongation reaction on *E. coli* 70S ribosomes (Fig. 5, Supplementary Fig. 5). As described below, finding of a DPR binding to the tunnel of a bacterial ribosome would enable a competition assay with the bacterial antibiotic

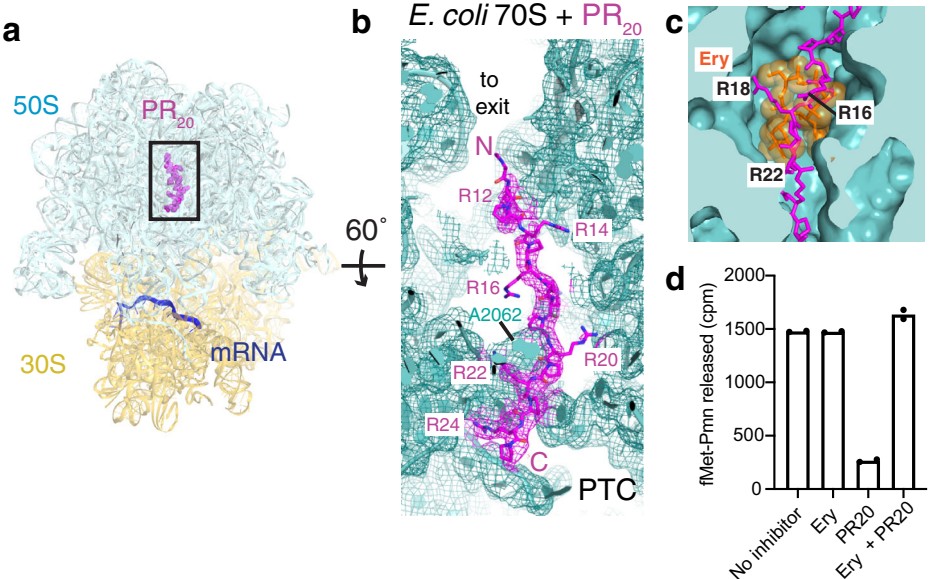

**Fig. 5 Cryo-EM shows that PR$_{20}$ occupies the polypeptide tunnel of the bacterial 70S ribosome. a** Overview of the structure of *E. coli* 70S•PR$_{20}$. **b** PR$_{20}$ binds to the 70S polypeptide exit channel. **c** PR$_{20}$ overlaps with the macrolide antibiotic binding site in the polypeptide tunnel. Erythromycin (Ery; orange) from the *E. coli* 70S•Ery crystal structure is shown (PDB:4V7U[87]). **d** Preincubation of ribosomes with erythromycin (Ery) relieves the inhibition of peptide bond formation by 2 μM PR$_{20}$, consistent with an overlapping binding site for Ery and PR$_{20}$. $n = 2$ independent experiments. Source data are provided as a Source Data file.

erythromycin to test the direct mechanism of PTC inhibition. We formed an initiation 70S complex with fMet-tRNA$^{fMet}$ and delivered Val-tRNA$^{Val}$ to the ribosomal A site by EF-Tu•GTP, in the presence or absence of PR$_{20}$. Maximum-likelihood classification of cryo-EM data resolved several elongation states of the 70S ribosome at up to 2.9-Å average resolution in the presence and absence of PR$_{20}$ (Supplementary Fig. 5b, c, j). The classification separated the empty ribosomes (no tRNA), from the substrate (vacant A site and filled P-site) and product (tRNA-dipeptide-bound A-site) states (Supplementary Fig. 5b, c).

Cryo-EM maps reveal PR$_{20}$ stably held at the constrictions of the conserved polypeptide tunnel, resembling the binding mode of this DPR protein to eukaryotic ribosomes (Fig. 5a, b). The poly-PR density was best resolved in the polypeptide tunnel of the ribosomes without tRNA (Fig. 5b, Supplementary Fig. 5c, d). In the presence of tRNA, however, the PR$_{20}$ density in the tunnel and tunnel constriction is very weak (Supplementary Fig. 5f, g), suggesting partial occupancy or a shift of poly-PR away from the P-site cleft and into the tunnel. Poly-PR displacement in reaction intermediates suggests that poly-PR may sample multiple registers along the polypeptide tunnel, slowing the elongation of nascent polypeptides towards the constriction. The higher abundance of vacant 70S with PR$_{20}$ (31.4%) rather than in its absence (20.4%) (Supplementary Fig. 5b, c) suggests that poly-PR may also destabilize the initiation 70S•fMet-tRNA$^{fMet}$ complex. In both the elongation and initiation complexes, strong density at the PTC suggests that poly-PR directly competes with tRNA binding to both the P and A site of the PTC.

Lastly, we tested whether the inhibition of peptidyl transfer is directly caused by binding of the DPR proteins to the polypeptide tunnel, rather than by a non-specific mechanism, *e.g.* binding of the positively-charged DPR proteins to tRNA or mRNA. We employed a competition assay, using the macrolide antibiotic erythromycin, an inhibitor of bacterial translation. Erythromycin binds at the constriction of the bacterial polypeptide tunnel near the PTC (Fig. 5c), but does not inhibit the puromycin reaction[59,60]. Thus, erythromycin is expected to restore peptidyl-transferase activity if it prevents poly-PR from binding the tunnel. We find that

preincubating 70S ribosomes with erythromycin prevents the inhibitory action of PR$_{20}$. In the presence of inhibitory concentrations of erythromycin and PR$_{20}$ the ribosome readily catalyzes the formation of the fMet-puromycin peptide bond (Fig. 5d). While this experiment cannot distinguish whether and how poly-PR perturbs the A-site and/or the P-site substrate (puromycin and fMet-tRNA$^{fMet}$, respectively), it supports the notion that peptidyl transfer is competitively inhibited by poly-PR binding to the ribosomal polypeptide tunnel.

## Discussion

This work uncovers how the toxic arginine-containing DPR proteins associated with *C9ORF72*-ALS/FTD inhibit translation. Our structures and biochemical results show that poly-PR and poly-GR with 20 or more dipeptide repeats strongly bind in trans to the polypeptide tunnels of all ribosome species we tested, from bacterial to mammalian (Supplementary Note 1 and Supplementary Fig. 6). Poly-PR and poly-GR binding to large subunits and assembled 80S ribosomes would compete with the binding of initiator Met-tRNA or elongator tRNAs, and strongly interfere with peptidyl transfer, consistent with translation initiation and elongation defects caused by these DPR proteins (this work and refs. [41,44,45]) (Fig. 6). By contrast, several other arginine-rich polypeptides, whose arginine content ranges from 46% to 100% (e.g. TAT (GRKKRRQRRRPPQ), FHV (RRRRNRTRRNRRRVR) and R$_{12}$ (RRRRRRRRRRRR)) have been shown not to cause translation inhibition or cell death even at 100 μM in concentration[45]. Poly-PR and poly-GR inhibition therefore results from strong and specific binding to the polypeptide tunnel via electrostatic and packing interactions, which are most pronounced for poly-PR (Figs. 4, 5). Our biochemical and structural findings are consistent with the higher cellular toxicity of poly-PR than poly-GR[20,22,36,39,45]. Our data are also consistent with the disease-causing length threshold of DPR proteins[61–63], revealing that the longer DPR proteins traverse the tunnel constrictions into the PTC, protruding into the tRNA binding sites. Stalling the arginine-containing DPR proteins in the tunnel can also explain

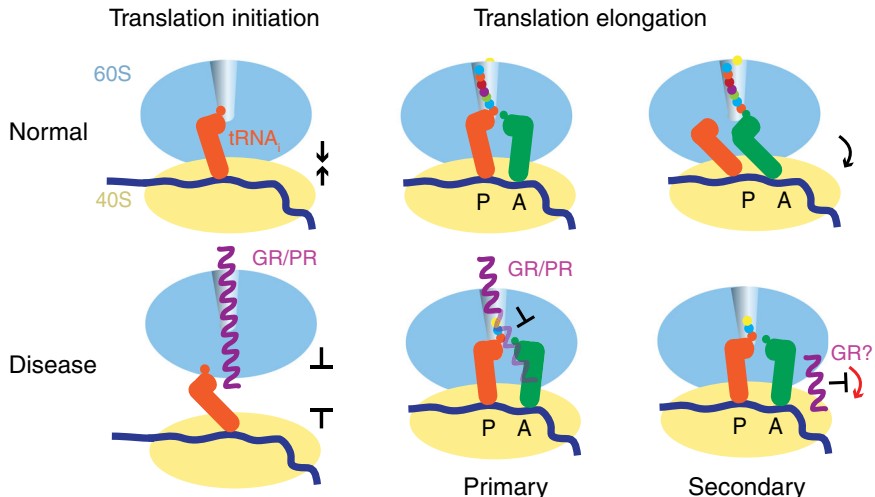

**Fig. 6 Mechanism of translation repression by poly-PR and poly-GR (purple): binding to the 60S (blue) polypeptide tunnel is consistent with perturbation of translation initiation (this work and ref. [44]) and binding to 80S inhibits translation elongation (this work and ref. [41]).** Poly-GR may also inhibit translation by binding to a secondary site at the interface of the 40S and 60S subunit thereby interfering with intersubunit rearrangements.

*in cis* translational stalling during DPR synthesis, which may disrupt cellular processes by triggering stress response (Supplementary Fig. 6b and refs. [19,40,46]). We also identified a putative secondary site at the intersubunit bridge B6, where poly-GR may bind directly or alter the tail of eL24. Here, poly-GR may interfere with intersubunit dynamics critical for the elongation step, consistent with poly-GR being predominantly an elongation inhibitor (Figs. [1]c, [2], and [6]) and at least partially accounting for differences with poly-PR, which appears to also inhibit initiating ribosomes (Fig. [2]). Our analyses were initially inspired by our earlier finding that poly-GR is preferentially associated with numerous mitochondrial and cytoplasmic ribosomal proteins in a co-immunoprecipitation experiment[24]. Remarkably, the mode of poly-PR and poly-GR action in the polypeptide tunnel as revealed by our current study resembles proline-rich antimicrobial peptides (PrAMPs), produced by eukaryotes as an antimicrobial strategy[64]. The PrAMPs share a conserved PRP repeat sometimes expanded to $(XR)_4$, which binds at the polypeptide tunnel similarly to poly-PR and poly-GR in our work (Supplementary Fig. 7a–d). PrAMPs also inhibit translation and compete with macrolide antibiotics for binding[65–69], echoing our findings with erythromycin (Fig. [5]d). The ability of mammalian cells to express PrAMPs[69] demonstrates that the $(XR)_n$ repeat sequences can be produced by cytoplasmic ribosomes and then interfere with translation in trans.

Indeed, cellular co-localization of the arginine-containing DPR proteins with nucleoli[20,22,39] and mitochondrial components[24,39,40], and the extracellular localization of these DPR proteins[22] demonstrate that *C9ORF72*-ALS/FTD-associated poly-PR and poly-GR can be liberated from ribosomes into the cytoplasm and organelles, and can impair global translation in trans. The inefficient translation required to make DPR proteins (due to *in cis* translation inhibition, Supplementary Note 1, Supplementary Fig. 6b and refs. [19,40,46]) and/or efficient protein degradation pathways may delay their accumulation to toxic levels, in keeping with the age-dependent late onset of ALS and FTD. Although concentrations of soluble DPR proteins in individual patient neurons are not known and difficult to measure, we speculate that accumulation of poly-GR and/or poly-PR in some neurons may be facilitated by stress conditions and/or processes associated with aging. The accumulating DPR proteins likely bind the ribosomal subunits and ribosomes, and inhibit translation, eventually leading to neuronal cell death.

Our work does not exclude other—ribosome-independent—mechanisms that may contribute to *C9ORF72*-ALS/FTD, including: defective nuclear transport[31,32], defective splicing[20,33], aggregation of non-arginine DPR proteins, such as poly-GA[34], repeat-RNA toxicity, or loss of functional *C9ORF72* protein, etc. (as reviewed in refs. [7,35]). Yet, concrete structural mechanisms of macromolecular inhibition for these possible pathways remain to be determined. Our findings of strong atomic interactions between the arginine-containing DPR proteins and the ribosome, and impairment of the central ribosome function, suggest a direct structural mechanism for cellular toxicity in *C9ORF72*-ALS/FTD and possibly in spinocerebellar ataxia type 36 (SCA36) where accumulation of poly-PR has recently been observed[70].

## Methods

$GR_{10}$, $GR_{20}$, $GR_{40}$, $PR_4$, $PR_{10}$, $PR_{20}$, and $PR_{40}$ peptides were synthesized and purified by CS Bio Co (Menlo Park, CA). Mass-spectrometry and HPLC were used to confirm the length and purity of peptides. Solid peptides were dissolved in water yielding stock concentrations between 1 and 10 mM, which were further diluted to specific concentrations for biochemical and cryo-EM experiments.

**Luciferase assays in rabbit reticulocyte lysates**. Rabbit reticulocyte lysates (RRL, Promega) were used to investigate the inhibition of translation by DPR proteins. We independently monitored translation of two mRNAs encoding non-homologous proteins of different lengths: nanoluciferase[71] and firefly luciferase. To measure the IC50s of the translational inhibitors cycloheximide (Akros Organics), harringtonine (AbCam), GDPCP (Jena Biosciences), $PR_{20}$ and $GR_{20}$, we monitored the translation of nanoluciferase mRNA in nuclease-treated RRL (Promega). In these experiments, 50% (final concentration) nuclease-treated RRL (Promega) was supplemented with 0.02 mM amino acids (Promega), 1% nanoluciferase substrate (Promega), and the appropriate concentration of each translation inhibitor for 5 min at 30 degrees. Next, 10 ng/µl (final concentration) 6xhis-Nanoluciferase mRNA flanked with the 5'- and 3'-UTR from rabbit beta-globin[48] was added and the luminescence reactions were continuously measured by an Infinite m1000 pro microplate reader (Tecan) for 15 min at 30 °C. The 1st derivative of the luminescence reactions was determined in Prism 8 (GraphPad Software, LLC) and the maximum was identified and the mean of three maximums from independent experiments for each inhibitor was taken as the rate. The rates were plotted versus inhibitors concentration, and the plots were fitted in Prism 8 (GraphPad Software, LLC) to a 3-parameter inhibition model ($Y =$ baseline $+$ (max-baseline)/$(1 + X/IC50)$) to obtain the IC50 values where the baseline was constrained to 0. This procedure was modified to separate translation initiation from translation elongation by changing the time of addition of the inhibitors to ~300 s after mRNA addition when RLUs had reached a value of ~50,000. Here, the maximum of the first derivative was taken in the 2nd or later measurement taken after inhibitor addition. In parallel, IC50 measurements were performed using 33% RRL (conditions), resulting in lower IC50 values than in the 50% RRL reaction, as expected

(325 nM (PR$_{20}$) and 425 nM (GR$_{20}$) at 33% vs 0.84 μM (PR$_{20}$) and 0.64 μM (GR$_{20}$) at 50% RRL).

Firefly luciferase translation was performed similarly to that for nanoluciferase, except that the efficiency of translation of 100 ng/μl firefly luciferase mRNA (Promega) in RRL was measured (in relative luminescence units, RLU) after 1 h of reaction incubation at 30 °C, in the absence or presence of PR$_{20}$, GR$_{20}$ and GP$_{20}$ (control) (Supplementary Fig. 1a–c).

The following additional controls indicate that DPR-induced changes in fluorescence were caused by translation inhibition. First, translation was not inhibited by 100 μM L-arginine, indicating that covalent linkages between arginines are required (Supplementary Fig. 1d). Second, poly-PR and poly-GR do not disrupt luciferase or nanoluciferase activity, when added after the luciferases have been translated in the absence of poly-PR and poly-GR (Supplementary Fig. 1e, f). Finally, they do not aggregate mRNAs at the inhibitory concentrations as described below (Supplementary Fig. 1g).

**Luciferase assays in bacterial cell extracts**. *E. coli* cell lysates (NEBExpress, NEB) were used to investigate bacterial translation inhibition by GR$_{20}$ and PR$_{20}$. We monitored the translation of an in vitro transcribed mRNA encoding a Shine-Dalgarno sequence preceding the nanoluciferase (Promega) open reading frame. The mRNA was prepared by in vitro transcription of a PCR-template obtained by amplifying the nanoluciferase open reading frame from the plasmid described in reference [48] with the following primers: T7 + SD-forward: TTTTTTAATAC GACTCACTATAGAAGAAGGAGATATACCATGGGCTCGAGCGGCGTCT TC, based on the pet28a+ plasmid and reverse: GCAATGAAATAAATTTC CTTTTATTAGCC. Eight microliter aliquots of the reaction mixture containing 30% NEBExpress S30 Synthesis Extract (NEB), 1.2x NEB Protein Synthesis Buffer, 1U/μl Murine RNAse Inhibitor (NEB), 1.25% Nanoluc substrate (Promega) was placed on a 384-well plate (Nunc, white). Then 1 μl of water or different concentrations (10–100 μM) of PR$_{20}$ or GR$_{20}$ or 200 μM erythromycin were added and the plate was incubated at 37 °C for 5 min inside of an Infinite m1000 pro microplate reader (Tecan). Next, the nanoluciferase mRNA was added to the final concentration of 20 ng/μl in a 10-μl reaction. Translation reactions were incubated at 37 °C for 20 min, luminescence of the samples was continuously measured by the microplate reader to record the time course of protein synthesis. Time progress curves, obtained in the experiments, were processed in MS Excel, and the rate of change of the relative light units between 100 and 200 s was measured, the mean and SEM of three experiments is reported in column graph in Supplementary Fig. 2a.

**Measurement of DPR-induced RNA aggregation**. To test whether arginine-containing DPR proteins cause mRNA aggregation at the DPR concentrations sufficient to inhibit protein synthesis (0.2 and 1 μM), we measured turbidity (optical density at 600 nm; OD600; similarly to [41]) of the solutions containing 100 ng/μl total HEK293 RNA. We incubated RNA with PR$_{20}$, GR$_{20}$, GP$_{20}$ (control), or water for 20 min at room temperature. Addition of PR$_{20}$ and GR$_{20}$ at concentrations that inhibit RRL translation and peptidyl transfer (from 0.2 to 1 μM) did not increase optical density relative to RNA alone, nor did water or GP$_{20}$. 10 μM PR$_{20}$ and GR$_{20}$ did not induce visible opalescence but resulted in increased OD600 (Supplementary Fig. 1g). Optical density was measured using NanoDrop One (Thermo Scientific, Waltham, MA, USA).

**Kinetic peptidyl transfer assays**. The effects of GR$_n$ or PR$_n$ on peptide bond formation in *E. coli* ribosome were tested in an in vitro assay that reacts the 70S ribosome loaded with [³⁵S]-fMet-tRNA$^{fMet}$ in the P-site with puromycin, an A-site tRNA mimic, and measures the release of the resulting dipeptide containing formyl-methionine covalently bonded to puromycin, as described[72]. *E. coli* tRNA$^{fMet}$ (Chemical Block) was aminoacylated with [³⁵S]-N-formyl-L-methionine (Perkin Elmer) as described[73]. 70S ribosomes were prepared from *E. coli* (MRE600) as described[72]. 70S ribosomes were stored in the ribosome-storage buffer (20 mM Tris-HCl, pH 7.0; 100 mM NH$_4$Cl; 20 mM MgCl$_2$) at −80 °C. A model mRNA fragment, containing the Shine-Dalgarno sequence and a spacer to position the AUG codon in the P-site and the UUC codon in the A site (GGC AAG GAG GUA AAA AUG UUC AAAAAA), was synthesized by IDT. 70S•mRNA•fMet-tRNA$^{fMet}$ complexes with GR$_n$ (where $n = 4$, 10 or 20) or PR$_n$ (where $n = 4$, 10, 20 or 40) were formed as follows. 330 nM 70S ribosomes were incubated with 11 μM mRNA in buffer containing 20 mM Tris-acetate (pH 7.0), 100 mM NH$_4$OAc, 12 mM Mg(OAc)$_2$, for 2 min at 37 °C. 730 nM [³⁵S]-fMet-tRNA$^{fMet}$ was added and incubated for 5 min at 37 °C. The solution was diluted 9.6-fold with the same buffer, followed by addition of a GR$_n$ or PR$_n$ repeat peptide (22x relative to its final concentration). The complex was incubated for 5 min at 37 °C prior to addition of puromycin dissolved in the same buffer (see below).

To test the effects of PR$_{20}$ on the mammalian peptidyl transferase, 80S ribosomes were purified from rabbit reticulocyte lysate (Green Hectares) and dissociated into 40S and 60S subunits as described in the section entitled "Cryo-EM complex of mammalian 80S ribosome with PR$_{20}$". A model leaderless mRNA fragment placing the AUG codon in the P-site and the UUC codon in the A site (CCAC AUG UUC CCCCCCCCCCCCCCCCCC), was synthesized by IDT. The 80S•mRNA•fMet-tRNA$^{fMet}$ complex with PR$_n$ (where $n = 10$, 20, or 40) or GR$_{20}$ was assembled as follows. one micrometer 60S subunit was mixed with the DPR

(starting with the initial DPR concentration of 33x relative to its final concentration in the reaction) in Assembly Buffer containing 20 mM Tris-acetate (pH 7.0), 100 mM KOAc, 10 mM Mg(OAc)$_2$, and the mixture was incubated for 5 min at 30 °C. In a separate microcentrifuge tube, 600 nM 40S was mixed with 16.5 μM mRNA in the same buffer and incubated for 2 min at 30 °C. 1095 nM [³⁵S]-fMet-tRNA$^{fMet}$ and the 60S solution were added (0.5 volumes of the 40S solution), resulting in the following concentrations: 330 nM 60S, 400 nM 40S, 730 nM tRNA, 11 μM mRNA and 11x of final concentration of a DPR in the buffer (20 mM Tris-acetate (pH 7.0), 100 mM KOAc, 10 mM Mg(OAc)$_2$). The solution was incubated for 5 min at 30 °C. The solution was diluted ten times with the same buffer, then puromycin (prepared in the same buffer) was added and time progress curves were recorded.

The kinetics of puromycin reaction on the 70S or 80S ribosomes were recorded and analyzed essentially as described[53]. An aliquot (4.5 μl) of the complex prior to addition of puromycin was quenched in 30 μl of saturated MgSO$_4$ in 0.1 M NaOAc to represent the zero-time point. Five microliters of 500 μM puromycin (in the same buffer as that used for the ribosomal complex) was added to 45 μl of the complex to initiate the reaction, yielding estimated final concentrations: 30 nM for the ribosome, 1 μM mRNA, 66 nM [³⁵S]-fMet-tRNA$^{fMet}$, 50 μM puromycin, and varying concentrations of PR$_n$ and GR$_n$ peptides, from 30 nM to 2 μM. After 6, 15, 30, 90 s, and 5, 15, 30, 60, and (for slower reactions) 180 min, 5-μl aliquots were quenched in 30 μl of saturated MgSO$_4$ in 0.1 M NaOAc. All quenched samples were extracted with 700 μl ethyl-acetate. 600 μl of the extract were mixed with 3.5 ml of Econo-Safe scintillation cocktail (RPI), and the amount of released [³⁵S]-labeled dipeptide was measured for each time point using a scintillation counter (Beckman Coulter, Inc.).

All time progress curves were recorded in duplicates, using independently prepared 70S or 80S complexes. Reaction rates were calculated using single-exponential or double-exponential (80S complexes) regressions for different concentrations of PR$_n$ and GR$_n$ using GraphPad Prism 8. Hyperbola fitting in Gnuplot was used to derive apparent $K_i$-values.

**Competition assay using erythromycin**. The assay was performed as described above for the puromycin kinetics, except that the 70S ribosomes were pre-incubated with 22 μM erythromycin for 5 min prior to adding PR$_{20}$. The stock solution of erythromycin contained 2 mM erythromycin in ethanol, so the control reaction (70S with no erythromycin) and the reactions with erythromycin were prepared to contain the same amount of ethanol (1%). The final concentrations after addition of puromycin were: 30 nM 70S, 66 nM [³⁵S]-fMet-tRNA$^{fMet}$, 1 μM mRNA, 2 μM PR$_{20}$ (no PR$_{20}$ in a control reaction), 20 μM erythromycin (or no erythromycin in a control reaction), 50 μM puromycin (in 20 mM Tris-acetate (pH 7.0), 100 mM NH$_4$OAc, 12 mM Mg(OAc)$_2$ and 1% ethanol). Samples were quenched at 300 s when the uninhibited reaction had reached a plateau.

**Polysome profiling**. Polysome profiles were obtained using RRL untreated with nuclease, so that it contained endogenous mRNA. 100 μl reactions containing 50% RRL (Green Hectare) 30 μM hemin (Sigma), 75 mM KCl, 0.5 mM MgCl$_2$, 1 mM ATP, 0.2 mM GTP, 2.1 mg/ml creatine phosphate, 30 μM of L-amino acid mix, 15 mg/ml yeast tRNA, 6 mM BME and, when appropriate, translation inhibitors (4 μM PR$_{20}$, 4 μM GR$_{20}$, cycloheximide, 40 μM harringtonine, or 1 mM GDPCP with 1 mM MgCl$_2$) were assembled on ice. To start translation, the reactions were warmed in a heat block at either 30 °C or 37 °C as noted. After 5 min, reactions were terminated by addition of 100 μl of ice-cold Stop Buffer (50 mM Hepes-KOH pH 7.4, 150 mM KOAc, 10 mM Mg(OAc)$_2$, and 200 μg/ml cycloheximide). The stopped reactions were applied to 10–50% sucrose gradients in Gradient Buffer (50 mM Hepes-KOH pH 7.4, 150 mM KOAc, 10 mM Mg(OAc)$_2$, 2 mM DTT, and 100 μg/ml cycloheximide) and spun in an SW-41 rotor (Beckman Coulter) at 197,568xg (avg) for 2 h at 4 °C. Gradients were pumped on a Gradient Station (BioComp Instruments) at a pump rate of 0.29 mm/min and the absorbance at 254 nm was measured by an Econo UV monitor (Biorad) zeroed with 10% sucrose-gradient buffer. The traces were output as CSV files and imported into GraphPad Prism, where the 80S peaks were aligned. Controls were highly reproducible day-to-day. Each experimental condition is shown compared to concurrent, uninhibited control.

**Toeprinting analyses of ribosome complexes**. Toeprinting analyses were carried out similarly to those in refs. [51,74], using 80S and polysome (4–5-some) fractions collected from polysome profiles as described in "Polysome Profiles" except that reactions were doubled in size (200 μL), performed at 37 °C, and were stopped with ice-cold 1xAMV Buffer (50 mM Tris-HCl pH 8.3, 50 mM KCl, 5 mM MgCl$_2$) with 200 μg/ml cycloheximide before being separated on at 10–35% sucrose gradient in 1xAMV buffer with 0.5 mM spermidine and 1 mM DTT. 600 μl of each fraction (80S or 4–5-mer) was supplemented with 10 mM DTT. Then 10 U of AMV reverse transcriptase (Promega), 120 nmol dNTPs, and 120 pmol of a 5′[6-carboxy-fluorescein (FAM)]-labeled primer (IDT-DNA) were added. Two primers specific to the endogenous beta-globin mRNA in RRL were used in separate reactions: 5′-(FAM)-GGACTCGAAGAACCTCTG (as in ref. [74]) annealed 136 nucleotides downstream of the AUG or 5′-(FAM)-TGCAATGAAAATAAATTTCCT annealed to the end of the 3′-UTR. Primers were extended at 37 °C for 30 min, and then the

**9**

reaction were extracted in 1:1 phenol:chloroform, supplemented with 1/10 volume of 3 M sodium acetate pH 5.2, and precipitated with 2.5 volumes of 100% ethanol. Pellets were resuspended in 10 µl of deionized formamide (Applied Biosystems) with 1/20 volume of a 35-500 nucleotide orange-dye labeled DNA size standard (MCLab) and loaded onto a 96-well plate. The samples were sent for fragment analysis by capillary electrophoresis to the UMass Chan Medical School Molecular Biology Core Laboratory using an ABI3730XL DNA Fragment Analyzer with capillaries loaded with POP-7 polymer and with an injection voltage of 12.5 kV. Data collection was performed with GeneMapper software (ABI). Data were exported as.fsa files and were analyzed (aligning to ladder and background correction) in GeneMarker 3.0.1 software (SoftGenetics) running in demo mode. Screen captures of traces were taken and traces were integrated using the "Quantification" feature and manually adjusting sliders to include the desired nucleotide range.

**Cryo-EM complexes of yeast 80S ribosomes with PR$_{20}$ or GR$_{20}$.** *Saccharomyces cerevisiae* 80S ribosomes complexes with PR$_{20}$ or GR$_{20}$, were assembled in vitro as follows. *S. cerevisiae* 40S and 60S ribosomal subunits were purified from strain W303 and stored in 1x Reassociation buffer 50 mM Tris-HCl pH 7.5, 20 mM MgCl$_2$, 100 mM KCl, 2 mM DTT, as previously described[75]. To form the 80S complex, 12 pmol of 40S ribosomal subunits were incubated in 1x Reassociation buffer with 240 pmol of a custom RNA oligonucleotide (Integrated DNA Technologies, Inc.) encoding the Kozak sequence, the start codon AUG, and an open reading (CCAC-AUG-UUC-CCC-CCC-CCC-CCC-CCC-CCC) and 60 pmol of tRNA$^{fMet}$ (ChemBlock) for 5 min at room temperature. Subsequently, 14.3 pmol of 60S ribosomal subunits were added and incubated for a further 5 min at room temperature. PR$_{20}$ or GR$_{20}$ were diluted to 30 µM in buffer B: 50 mM Tris-HCl pH 7.5, 10 mM MgCl$_2$, 100 mM KCl and mixed 1:1 with the 80S assembly reaction for a final reaction containing: 300 nM 40S, 360 nM 60S, 6 µM mRNA, 1.5 µM tRNA$^{fMet}$, and 15 µM PR$_{20}$. Complexes were incubated for a further 5–10 min at room temperature prior to plunging cryo-EM grids.

Quantifoil R2/1–4 C grids coated with a 2-nm thin layer of carbon were purchased from EMSDiasum. The grids were glow discharged with 20 mA current with negative polarity for 60 s in a PELCO easiGlow glow discharge unit. A Vitrobot Mark IV (ThermoFisher Scientific) was pre-equilibrated to room temperature and 95% humidity. 2–3 µl of the 80S assembly reaction was applied to the grid, incubated for 10–20 s, blotted for 5 s, and then plunged into liquid-nitrogen-cooled liquid ethane.

**Cryo-EM complex of mammalian 80S ribosome with PR$_{20}$ or GR$_{20}$.** *Oryctolagus cuniculus* 80S ribosomal complexes with buffer, PR$_{20}$ or GR$_{20}$ were prepared using ribosomal subunits purified from rabbit reticulocyte lysate (RRL), based on the procedures described in references[76,77]. 60 ml of RRL (Green Hectares) was thawed and mixed 1:1 in 2x RRL dissolving buffer (10 mM HEPES-KOH pH 7.1, 30 mM KCl, 22 mM Mg(OAc)$_2$, 2 mM EDTA, 4 mM DTT, 0.6 mg/ml heparin, and Complete Protease Inhibitor (Roche)). The solution was split and layered over four sucrose cushions of 20 mM Bis-Tris pH 5.9, 300 mM KCl, 200 mM NH$_4$Cl, 10 mM Mg(OAc)$_2$, 30% (w/v) sucrose and 5 mM DTT and centrifuged in an Optima XPN ultracentrifuge (Beckman Coulter) in a Type 45Ti rotor for 16 h at 144,962xg (avg). The crude ribosome pellets were gently resuspended in re-dissolving buffer (20 mM Tris-HCl pH 7.5, 50 mM KCl, 4 mM Mg(OAc)$_2$, 1 mM EDTA, 2 mM DTT, 1 mg/ml heparin, and Complete Protease Inhibitor (Roche)). To disrupt polysomes, 1 mM puromycin was added to ribosome solution and incubated 20 min at 37 °C, 20 min at room temperature, and 20 min on ice. To separate subunits, the KCl concentration in the ribosome solution was gradually adjusted from 50 mM to 500 mM followed by a 30-min incubation at 4 °C. The ribosomal subunits were separated using a 10–35% sucrose gradient (20 mM Tris-HCl pH 7.5, 500 mM KCl, 4 mM Mg(OAc)$_2$, 2 mM DTT). 12 tubes were centrifuged for 16 h at 64,047xg (avg) in SW28 rotors in an Optima XPN ultracentrifuge (Beckman Coulter), and the fractions for the 40S and 60S peaks were collected using a Gradient Master (Biocomp Instruments). The 40S and 60S subunits were concentrated and exchanged into Storage Buffer (20 mM Tris-HCl pH 7.5, 100 mM KCl, 2.5 mM Mg(OAc)$_2$, 2 mM DTT, 0.25 M sucrose) using centrifugal filters (100 kDa MWCO; Amicon).

The complexes of rabbit 80S ribosomes with buffer, PR$_{20}$ or GR$_{20}$ for cryo-EM were prepared similarly to that for puromycin release assays. 1 µM rabbit 60 S subunits were pre-incubated for 5 min at 30 °C with 10 µM PR$_{20}$ or 10 µM GR$_{20}$ or an equivalent volume of water. The assembly was carried out in Assembly Buffer (20 mM Tris-acetate (pH 7.0), 100 mM KOAc, 10 mM Mg(OAc)$_2$). Meanwhile, 40S subunits were preincubated with mRNA for 5 min at 30 °C in the same buffer. tRNA$^{fMet}$ (Chemical Block) was added to 40S such that final concentrations were 1.2 µM 40S, 40 µM mRNA, and 2.4 µM tRNA$^{fMet}$. Then the 40S and 60S reactions were mixed 1:1 and incubated for 5 min at 30 °C. Finally, in optimizing the ribosome density on the grids, we found it appropriate to dilute the reaction 60% with Assembly Buffer with or without additional PR$_{20}$ or GR$_{20}$. The final 80S assembly reactions used for data collection contained: 200 nM 60S, 240 nM 40S, 8 µM mRNA, 480 nM tRNA$^{fMet}$, and either 0 or 5 µM DPR (PR$_{20}$ or GR$_{20}$).

For PR$_{20}$ complex, 400 M copper grids coated with lacey carbon and a 2-nm thin layer of carbon (Ted Pella Inc.) were glow discharged with 20 mA current with negative polarity for 60 s in a PELCO easiGlow glow discharge unit. For buffer or GR$_{20}$ complex, Quantifoil R2/1 holey-carbon grids coated with a thin layer of

carbon (EMSDiasum). In all three cases, Vitrobot Mark IV (ThermoFisher Scientific) was pre-equilibrated to 4 °C and 95% relative humidity. 2.5 µl of the 80S assembly reaction was applied to grid, incubated for 10 s, blotted for 4 s, and then plunged into liquid-nitrogen-cooled liquid ethane.

**Cryo-EM complex of bacterial 70S ribosome with PR$_{20}$.** 30S and 50S ribosomal subunits were prepared from MRE600 *E. coli* as described[72] and stored in Buffer A (20 mM Tris-HCl, pH 7, 10.5 mM MgCl$_2$, 100 mM NH$_4$Cl, 0.5 mM EDTA, 6 mM β-mercaptoethanol) at −80 °C. *E. coli* EF-Tu, fMet-tRNA$^{fMet}$ and Val-tRNA$^{Val}$ (tRNA$^{Val}$ from Chemical Block) were prepared as described[78]. mRNA containing the Shine-Dalgarno sequence and a linker to place the AUG codon in the P-site and the Val codon (cognate complex) in the A site was synthesized by Integrated DNA Technologies Inc (GGC AAG GAG GUA AAA <u>AUG GUA</u> AGU UCU AAA AAA AAA AAA).

The 70S complexes were prepared as follows. Heat-activated (42 °C, 5 min) 30S ribosomal subunits (1 µM) were mixed with 50S ribosomal subunits (1 µM) and with mRNA (4 µM) (all final concentrations) in Reaction buffer (20 mM HEPES•KOH, pH 7.5, 20 mM MgCl$_2$, 120 mM NH$_4$Cl, 2 mM spermidine, 0.05 mM spermine, 2 mM β-mercaptoethanol) for 10 min at 37 °C. Equimolar fMet-tRNA$^{fMet}$ was added to the ribosomal subunits and incubated for 3 min at 37 °C. Subsequently, 15 µM PR$_{20}$ was added and incubated another 5 min at 37 °C, then the reaction was held on ice. Concurrently, the ternary complex of Val-tRNA$^{Val}$•EF-Tu•GTP was prepared as follows. 6 µM EF-Tu was pre-incubated with 1 mM GTP (Roche) in Reaction buffer for 5 min at 37 °C and then was supplemented with 4 µM Val-tRNA$^{Val}$ (all final concentrations). After an additional minute at 37 °C, the ternary complex reaction was also kept on ice to prepare for plunging. The 70S reaction and ternary complex was mixed in 1x Reaction buffer and incubated at 37 °C for 10 min for the elongation step to complete. The final reaction had the following concentrations: 330 nM 50S; 330 nM 30S; 1.3 µM mRNA; 330 nM fMet-tRNA$^{fMet}$; 500 nM EF-Tu; 83 µM GTP, 330 nM Val-tRNA$^{Val}$ with or without 5 µM PR$_{20}$.

Quantifoil R2/1 holey-carbon grids coated with a thin layer of carbon (EMSDiasum) were glow discharged with 20 mA current with negative polarity for 60 s in a PELCO easiGlow glow discharge unit. A Vitrobot Mark IV (ThermoFisher Scientific) was pre-equilibrated to 5 °C and 95% relative humidity. 2.5 µl of the ribosomal complex prepared with 5 µM PR$_{20}$ were applied to chilled grids, and then blotted for 4 s prior to plunging into liquid-nitrogen-cooled liquid ethane.

**Electron microscopy.** Data for the *S. cerevisae* 80S•tRNA•PR$_{20}$ or *S. cerevisae* 80S•tRNA•GR$_{20}$ were collected on a Titan Krios electron microscope (ThermoFisher Scientific) operating at 300 kV and equipped with a Gatan Image Filter (slit width 20 eV) (Gatan Inc.) and a K2 Summit direct electron (Gatan Inc.) targeting 0.5–2.0-µm underfocus. For 80S•tRNA•PR$_{20}$, a dataset of 203,089 particles from 3033 movies was collected automatically using SerialEM[79] using beam tilt to collected five movies per hole at four holes between stage movements[80]. The movies had a total of 30 frames with 1 e$^−$/Å$^2$ per frame for a total dose of 30 e$^−$/Å$^2$ on the sample. The 80S•tRNA•GR$_{20}$ dataset had 467,615 particles from 3451 movies collected using beam tilt to collected six shots per hole. 2071 movies were used for data analysis after exclusion of suboptimal movies (due to poor contrast, ice, or collection outside of desired area). The movies had a total of 32 frames with 0.9 e$^−$/Å$^2$ per frame for a total dose of 30 e$^−$/Å$^2$ on the sample. The super-resolution pixel size was 0.5294 Å for both datasets.

Data for the rabbit 80S•tRNA and rabbit 80S•tRNA•GR$_{20}$ ribosomal complexes were collected on a Titan Krios electron microscope (ThermoFisher Scientific) operating at 300 kV and equipped with a Gatan Image Filter (slit width 20 eV) (Gatan Inc.) and a K3 Summit direct electron (Gatan Inc.) targeting 0.5–2.0-µm underfocus. For the rabbit 80S, a dataset of 54,531 particles from 1099 movies was collected automatically using SerialEM[79] using beam tilt to collected five movies per hole at four holes between stage movements[80]. The movies had a total of 30 frames with 1 e$^−$/Å$^2$ per frame for a total dose of 30 e$^−$/Å$^2$ on the sample. The 80S•tRNA•GR$_{20}$ dataset had 276,778 particles from 5320 movies. The super-resolution pixel size was 0.415 Å for both datasets.

Data for two complexes—(1) rabbit 80S•tRNA•PR$_{20}$ (2) *E. coli* 70S•tRNA•PR$_{20}$—were collected on a Talos electron microscope (ThermoFisher Scientific) operating at 200 KV and equipped with a K3 direct electron detector (Gatan Inc.) targeting 0.6–1.8-µm underfocus. Data collection was automated using SerialEM[79] using beam tilt to collect multiple movies (e.g., four movies per hole at four holes) at each stage position[80]. The rabbit 80S•tRNA•PR$_{20}$ dataset had 3031 movies, at 19 frames per movie at 1.5 e$^−$/Å$^2$ per frame for a total dose of 30 e$^−$/Å$^2$ on the sample, yielding 246,885 particles. The *E. coli* 70S•PR$_{20}$ dataset had a total of 20 frames per movie, with 1.1 e$^−$/Å$^2$ per frame for a total dose of 36 e$^−$/Å$^2$ on the sample, comprising 1024 movies yielding 172,278 particles.

In all cases, movies were aligned on the fly during data collection using IMOD[81] to decompress frames, apply the gain reference, and to correct for image drift and particle damage and bin the super-resolution pixel by 2.

**Cryo-EM data classification and maps**

*Yeast 80S complexes.* Early steps of 3D map generation from CTF determination, reference-free particle picking, and stack creation were carried out in cisTEM,

while particle alignment and refinement was carried out in Frealign 9.11[82] and cisTEM[83]. To speed up processing, 2×-, 4×- and 6×-binned image stacks were prepared using resample.exe, which is part of the Frealign distribution[82]. The initial model for particle alignment of 80S maps was EMD-5976[84], which was down-sampled to match 6× binned image stack using EMAN2[85]. Three rounds of mode 3 search alignment to 20 Å were run using the 6× binned stack. Next, 25-30 rounds of mode 1 refinement were run with the 4×, 2× and eventually unbinned stack until the resolutions stopped improving (2.96 Å for PR20 and 3.06 Å for GR20), similarly to the "auto-refine" procedure in cisTEM. Next, one round of beam shift refinement and per-particle CTF refinement improvement the final maps to 2.41 Å (PR20) and 2.73 Å (GR20), which were used for model building. 3D maximum-likelihood classification into six classes was used to separate the 60S maps shown in Supplementary Fig. 3d, e.

To separate the ribosomes with and without DPR proteins in the tunnel, we used maximum-likelihood classification into up to six classes, applying a 30 Å focus mask around the center of the polypeptide tunnel of the 2× stack at the resolution of 6 Å. This strategy revealed features for poly-PR or poly-GR in all classes, but the density differed suggesting that heterogeneous conformations of the chains are possible in the tunnel.

*Rabbit 80S complexes.* CTF determination, micrograph screening to remove off-target shots, reference-free particle picking, and stack creation were carried out in cisTEM. Particle alignment and refinement were carried out in Frealign 9.11 and cisTEM. Box size for PR20 dataset was 608 × 608 × 608 while for the buffer and GR20 datasets a larger box, 720 × 720 × 720, was initially used. To speed up processing, binned image stacks (e.g. 8×, 4×, or 2×) were prepared using resample.exe. The initial model for particle alignment of 80S maps was EMD-4729 (80S•tRNA•PR20 complex) or start-up model generated in cisTEM from a subset of the 80 S•tRNA•GR20 dataset, which were down-sampled to match the most binned stack and low-pass filtered to 30 Å, using EMAN2. Two rounds of mode 3 search alignment to 20 Å were performed using the most binned stack. Next, rounds of mode 1 refinement were run with the less binned and eventually the unbinned stack as we gradually added resolution shells until the reconstruction stopped improving. For the overall Rabbit 80S•tRNA•GR20 map, the resolution reached 2.6 Å. For the overall rabbit 80S buffer control map, the resolution reached 2.9 A. Beam tilt or per-particle CTF refinement did not further improve the 80S•tRNA with buffer or 80S•tRNA•GR20. For the PR20 map, when resolution reached 3.50 Å a round of beam shift refinement improved the resolution to 3.11 Å and a round of per-particle CTF refinement further improved the resolution to 3.02 Å. 3D maximum-likelihood classification into 12 classes (using the high-resolution limit of 12 Å) was then used to resolve different ribosome states including the free 60S state (Supplementary Fig. 4). Similar classes were merged (using merge_classes.exe) as shown in Supplementary Fig. 4. To facilitate comparison between the three rabbit 80S datasets (buffer, GR20 and PR20) at similar σ levels (in Supplementary Fig. 4) and ease opening the maps in Pymol, the larger buffer and GR20 volumes were cropped to 640x640x640 using EMAN2[85] to match the dimensions in Å of the PR20 volumes, and the models were shifted correspondingly.

*70S complex.* Early steps of 3D map generation from CTF determination, reference-free particle picking, and stack creation were carried out in cisTEM, while particle alignment and refinement was carried out in Frealign 9.11 and cisTEM. To speed up processing, 2×- and 8×-binned image stacks were prepared using resample.exe. The initial model for particle alignment of 70S maps was the 11.5 Å map EMD-1003[86], which was down-sampled to match the 8× binned image stack using EMAN2. Three rounds of mode 3 search aligned to 20 Å were run using the 8× binned stack. Next, multiple rounds of mode 1 refinement were run with the 8×, 2× stacks, and eventually the unbinned stack, as we gradually added resolution shells (limit of 6 Å) and resolution was 2.98 Å (PR20). Next, beam shift refinement was used to improve the overall resolution to 2.86 Å, also resulting in improved map features for both rRNA and proteins.

3D maximum-likelihood classification was used to separate particles into classes of different compositions (Supplementary Fig. 5). Classes were added until classification revealed the hybrid 70S class with the P/E-tRNA and A/P-tRNA (the product of the dipeptide reaction). This processing required 12 classes for 60 rounds for the 70S•PR20 complex. PR20 classes were reconstructed with the original refined parameters (to 6 Å maximum resolution) and overall beam shift parameters to yield the final map for model building (obtained by merging four 70S classes without tRNA) or structural analyses (Supplementary Fig. 5c–i). Comparison of poly-PR and erythromycin binding sites in Fig. 5c was prepared by structural superimposition of the 23S rRNA of the 70S•tRNA•PR20 structure to that of *E. coli* ribosome bound with erythromycin[87].

### Model building and refinement

*Yeast 80S model building.* The 3.0-Å crystal structure of the *Saccharomyces cerevisiae* 80S ribosome (PDB: 4V88[58]) was used as a starting model for structure refinement. Because of the heterogeneity in the density for the 40S subunit conformation in our highest-resolution map, we only modeled and refined the 60S subunit and the tRNA bound to the L1 stalk. The model for the P/E-site tRNA^fMet and the L1 stalk was derived from the 6.2-Å cryo-EM structure of the

*Saccharomyces cerevisiae* 80S ribosome bound with 1 tRNA (PDB: 3J77[84]). PR20 was modeled in Coot (vs. 0.8.2)[88] in either C-out or N-out conformations starting from the PrAMPs Bac7 (PDB:5HAU[68]) or Api137 (PDB:5O2R[67]) and their fits into well-resolved density near residue A883. Local real-space refinement in Phenix[89] with and without Ramachandran restraints was used to determine the best directionality (N-out) for the PR chain in the polypeptide exit channel. The putative GR20 models were created in Coot (vs. 0.8.2), however, the density prevented unambiguous building of a single continuous model due to the high flexibility of the glycine-rich protein chain.

*Rabbit 80S model building.* Cryo-EM structures of XBP1u-paused *Oryctolagus cuniculus* ribosome-nascent chain complex omitting XBP1 and tRNA were used as starting models for structure refinement using PDB:6R6Q[90] (rotated state) for the 3.1-Å cryo-EM map of rabbit 80S with PR20 and PDB:6R5Q (non-rotated state) for the 2.9-Å cryo-EM map of the rabbit 80S with GR20, which showed a weak P-tRNA and non-rotated but more open 40S conformation. Domains (60S, 40S head and 40S body) of each 80S and the L1 stalk from PDB:6HCJ[91] were rigid-body fitted into each cryo-EM map. P-tRNA in the 80S•tRNA•GR20 structure was modeled from PDB: 5UYM[92] omitting the CCA end, which was disordered in the map. P/E-tRNA in the PR20 structure was modeled from PDB: 6WDF[92]. mRNA was rebuilt from PDB: 6R5Q to reflect the new sequence. PR20 from the yeast 80S-PR20 structure was remodeled in Coot (vs. 0.8.2) to fit the density in the rabbit 80S-PR20 map. GR20 was built into unmodeled density in the peptide tunnel using Coot (vs. 0.8.2), and GR20 backbone was built into unmodeled density near eL24.

*70S model building.* The 3.2-Å cryo-EM structure of 70S•tRNA•EF-Tu•GDPCP (PDB: 5UYM[78]) was used as a starting model for structure refinements with tRNAs and EF-Tu removed. The ribosome was fitted into maps using Chimera[93] with independent fitting for the 50S, L1 stalk, L11 stalk, 30S body, shoulder, and head. Local adjustments to nucleotides in the polypeptide exit channel and decoding center were made using Pymol[94]. PR20 was remodeled in Coot (vs. 0.8.2), starting from the N-out conformations from the yeast 80S-PR20 complex.

*Model refinement.* The models were refined using real-space simulated-annealing refinement using RSRef. [95,96] against corresponding maps. Refinement parameters, such as the relative weighting of stereochemical restraints and experimental energy term, were optimized to produce the optimal structure stereochemistry, real-space correlation coefficient and R-factor, which report on the fit of the model to the map[97]. The structures were next refined using phenix.real_space_refine[98] to optimize protein geometry and B-factors. The resulting structural models have good stereochemical parameters, characterized by low deviation from ideal bond lengths and angles and agree closely with the corresponding maps as indicated by high correlation coefficients and low real-space R factors (Supplementary Table 1). Figures were prepared in Pymol[94] and Chimera[93].

**Reporting summary**. Further information on research design is available in the Nature Research Reporting Summary linked to this article.

## Data availability
The models generated in this study have been deposited in the RCSB Protein Data Bank under the following accession codes: 7TOO (yeast ribosome with GR20), 7TOP (yeast ribosome with PR20), 7TOQ (rabbit ribosome with PR20), 7TOR (rabbit ribosome with GR20) and 7TOS (*E. coli* ribosome with PR20). The cryo-EM maps used to generate models in this study have been deposited in the Electron Microscopy Database under the following accession codes: EMD-26033 (yeast ribosome with GR20), EMD-26034 (yeast ribosome with PR20), EMD-26035 (rabbit ribosome with PR20), and EMD-26036 (rabbit ribosome with GR20) and EMD-26037 (*E. coli* ribosome with PR20). The coordinate files used in this study are available in the RCSB Protein Data Bank: 1M90 1VY5, 3JCT, 3J77, 4V7U, 4V88, 5APN, 5HAU, 5H4P, 5LZS, 5O2R, 5UYM, 6HCJ, 6FKR, 6R5Q, 6R6Q and 6WDF. The electron density maps used in this study are available from the Electron Microscopy Database: EMD-1003, EMD-4729, and EMD-5976. Source data are provided with this paper.

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

## Acknowledgements
We thank Chen Xu, KangKang Song, and Kyounghwan Lee for help with screening cryo-EM grids and for assistance with data collection at the UMass Chan Medical School cryo-EM facility; Christine Carbone for assistance with ribosome purification; Megan McNeil for assistance with structural modeling; Darryl Conte Jr. and members of the Korostelev lab for helpful comments on the manuscript. This study was supported by the Dan and Diane Riccio Fund for Neuroscience (to AAK and FBG), and by NIH Grants R01NS101986 and R37NS057553 (to FBG), and R01GM107465 and R35GM127094 (to AAK).

## Author contributions
Conceptualization: A.B.L, F.-B.G, and A.A.K; methodology: all including GD; validation: A.B.L and A.A.K, investigation: A.B.L., E.S., D.S., S.L., A.P., and S.Z.; resources: F.-B.G. and A.A.K. Writing—original Draft: A.B.L., F.-B.G., and A.A.K.; writing—review and editing: all; visualization:, A.B.L., D.S., S.L., and A.A.K. Supervision: F.-B.G. and A.A.K. Funding acquisition: F.-B.G. and A.A.K.

## Competing interests
The authors declare no competing interests.
