## [Peer Review File · Nature Communications]

Ribosome inhibition by C9ORF72-ALS/FTD-associated poly-PR and poly-GR proteins revealed by cryo-EMReviewers' Comments:

Reviewer #1:

Remarks to the Author:

The current manuscript by Loveland et al. addresses a question of how the expansion of the G4C2 repeats of the C9ORF72 gene results in amyotrophic lateral sclerosis (ALS). More specifically, the authors study how the encoded poly-PR and poly-GR gene products of these repeats inhibit protein synthesis by the eukaryotic ribosome. Undoubtedly, this topic is extremely important because accumulation of such polypeptides is proven to be the leading cause of many neurodegenerative disorders in humans. In this work, the authors provide biochemical evidence that poly-PR and poly-GR peptides inhibit protein translation by eukaryotic ribosome. Using competition with erythromycin, the authors also show that this inhibition is due to specific interactions of the peptide with the macrolide binding site located in the nascent peptide exit tunnel of the ribosome. Finally, based on several cryo-EM structures of ribosomes from yeast, rabbit, and E.coli in complex with poly-PR and poly-GR peptides the authors claim to uncover the details of the translation inhibition mechanism. The main conclusion from this work is that poly-PR/GR peptides bind to the ribosome exit tunnel thereby inhibiting protein synthesis.

Overall, this work is presented as a very concise (even too much concise at parts) and clearly written manuscript. One of the strong sides of this work is the combination of a structural approach with the biochemical techniques, which only to some extent reveal the molecular mechanism of translation inhibition by poly-PR/GR peptides. In summary, the results of this work could, in principle, be interesting to a broad readership of Nature Communication. However, there are several major concerns denoted below that need to be addressed by the authors.

Major issues:

1. One of the main conclusions of this work that poly-PR/GR peptides inhibit peptidyl transfer was made based on the results of puromycin assay on E.coli 70S and rabbit 80S ribosome. It is not surprising that binding of poly-PR/GR peptides to 70S/80S ribosomes decreases puromycin reactivity because these peptides simply block the access of puromycin to the A site of the ribosome. Therefore, puromycin assay, in this case, measures not the inhibition of peptidyl transfer reaction but rather the ability of puromycin to outcompete poly-PR/GR peptides from the ribosomal A site. In my opinion, this assay is inappropriate here because it does not allow us to discriminate between competition for the A site and the actual inhibition of PTC activity. Especially with the provided structures, it is pretty clear that these peptides prevent access of puromycin to the A site. Thus, it appears to be impossible to draw an unambiguous conclusion from the provided puromycin assay results.
2. The models of translation inhibition by poly-PR/GR peptides are based almost exclusively on the structural data. However, a definitive answer about the molecular mechanism of inhibition could be provided only by functional studies. In my opinion, this type of data to support the claims should be provided in work published in such a high-rank scientific journal. For example, a simple toe-printing experiment that takes less than a day can reveal which stage of translation is being inhibited by the poly-PR/GR peptides and whether the ribosome initiation complexes can be assembled in the presence of these peptides or not.
3. Based on some previously published data, the authors state that these peptides act in trans (a peptide is being synthesized on ribosome A but inhibits ribosome B), disfavoring the idea that they could also work in cis (inhibit and stall the very ribosome that synthesized them). Again, a simple toe-printing experiment can easily distinguish between these two possibilities. However, what I find even more important is that the suggested in trans mode of action of poly-PR/GR peptides is contradictory to their orientation in the ribosome. Most of the antibacterial PrAMP peptides that are very similar to the poly-PR/GR peptides studied here bind ribosomal tunnel in the reverse direction, with the C-terminus sticking into the tunnel. This explains how these PrAMPs could be synthesized by the

ribosome, in the first place, and then how they inhibit other ribosomes in trans. Because poly-PR/GR peptides bind ribosome with their N-termini facing the tunnel, it is unclear how they could even be synthesized without the same ribosome being stalled due to their inhibitory in cis action in this case?

4. How sure are the authors that they actually observe the charge density maps corresponding to the peptides in the tunnel of the yeast 80S-GR20 and rabbit 80S-PR20 structures shown in figures 2d and 2e, respectively? Undoubtedly, there is a clear density for the peptides in the tunnel in the case of yeast 80S-PR20 and Ecoli 70S-PR20 structures. However, the other two maps are questionable. For example, by closely looking at figure 2d, one can see that the putative density for the GR20 peptide covers only a small portion of the peptide, and the model does not even fit it well. Even worse is the density for the peptide shown in Figure 2e, where only a short fragment of density for the peptide could be observed next to the PTC. Moreover, on Figures 2b, 2d, and 2e, the map for the peptide is depicted at 2.5sigma, whereas the map for the ribosomal parts is shown at 4sigma. This makes the density for the peptide look stronger than, in fact, it is – which is inappropriate. The authors should either remove the density for the ribosome and show only the density for the peptides or show the density for all parts at the same uniform contour level. Furthermore, in Figure 2e, there is an additional map shown at 1.1sigma for the middle portion of the peptide in the tunnel, which is clearly invisible at 2.5sigma. This definitely misleads because it makes an impression as if the density for the peptide is observed throughout the tunnel up to the constriction point, while there is only a short density fragment next to the PTC.

Relatively minor issues:

5. The authors used L-arginine as a negative control for their translation inhibition studies. A control should be an experiment with only a single parameter altered at a time. Therefore, the authors should have considered using poly-R, poly-G, and poly-P peptides of the same length.

6. The authors state that they checked the effect of poly-PR/GR peptides on puromycin reactivity with the 70S ribosomes from E.coli because these peptides were known to interfere with mitochondrial translation. I need to say that, maybe using the E.coli 70S ribosome as a model for mammalian mitochondrial 55S ribosomes would have been appropriate 30-40 years ago, it is not really the case anymore because these ribosomes are drastically different as demonstrated by numerous studies. Therefore, if the authors would like to make a claim about inhibition of mitochondrial ribosomes by the poly-PR/GR peptides, they should really use isolated mitochondrial ribosomes, not the bacterial ones.

7. It is very confusing that the results of the puromycin assay with the 70S and 80S ribosomes shown in Figures 1e and 1f, respectively, are shown using completely different representations of basically the same assay. These two panels should have the same style of representation.

8. Although the authors claim that poly-PR/GR peptides are oriented with their N-termini towards the exit, there is no evidence in the manuscript to support this conclusion. Given that there are only short fragments of these longer peptide repeats visible in the tunnel, how is it even possible to determine such orientation? Is it possible that poly-PR/GR peptides bind ribosome in a reverse orientation similar to PrAMPs?

9. The experiment with erythromycin and E.coli 70S ribosomes was accomplished to show that poly-PR/GR peptides inhibit translation by binding at the specific site in the ribosomal tunnel and also to show that erythromycin and poly-PR/GR peptides have overlapping binding sites. But isn't this totally obvious from the structural studies presented here?

10. The figures showing the cryo-EM density maps should be made significantly bigger to allow the reader to examine the map without a magnifying glass.

11. The authors provided almost no introduction at the beginning of the manuscript. Instead, they go

straight to the "G4C2 repeats in the C9ORF72". What are G4C2 repeats? What is C9ORF72? A few paragraphs spanning a page or two introducing this topic should be added to the manuscript in my opinion.

Reviewer #2:

Remarks to the Author:

This manuscript presents biochemical and structural insights into how expanded G4C2 repeats located in C9ORF72 inhibit translation. The repeat length in C9ORF72 correlates to ALS/FTD pathologies and this manuscript aims to identify the molecular basis for this function. Overall this is an interesting manuscript but it is hard to assess the data given the shortness of the manuscript. Many important details of prior data that rationalizes their own work is also missing. It is also confusing how the authors use multiple different model systems without clear rationales for their experiments. Together this constitutes a significant weakness because the absence of such details coupled with a lack of rationale made it difficult to determine the impact of this work as discussed below.

Finally, how do the repeats work? The authors perform all their assays with the impression that translated poly-PR and poly-GR can somehow rebind the ribosome after it has been translated and released. How it does this is unknown. The authors could perform biochemical assays to test this but do not. In other words, many cryo-EM structures are solved without a rationale and therefore the biological significance of some of these structures is minimal. Figures presented of the structures are not clear and one cannot discern the details the authors refer to in the text (Figure 2b-e; Figure 3b). Lastly, their model figure is not based upon the data presented or tested here and to my knowledge, not shown in other work. All of these issues are explained in more detail below.

Minimal information

-A number of the sections are incredibly short and lack details required to understand prior studies and the rationale and outcome of the experiments presented in this manuscript. For example, in the Introduction there are not enough details to understand previous experiments needed to know how to place these new experiments. For example, the authors describe experiments that demonstrate how poly-PR and poly-GR impair cellular translation but there are missing details. What system were these experiments performed in (animal model, ALS-FTD patients, cell culture etc)? Were these proteins overexpressed or do the authors mean that endogenous levels of these proteins/peptides alone affected translation? And finally what triggers the expression of these proteins in human patient brains? There is a sentence of how expression of poly-PR and poly-GR correlates with neurodegeneration but more information is needed to understand the precise correlation.

-When the authors describe how "poly-PR" and "poly-GR" impair global cellular translation, do they mean C9ORF72 proteins containing an expansion of these repeats or did these assays simply rely on peptides containing ONLY contain PR or GR? (first sentence of the Results section- this information though should go into the Intro). What other parts of the protein are the authors removing in these assays? This is important because the authors in this manuscript only use PR or GR containing peptides but they have not set out the premise that it is acceptable to use these peptides as mimics of C9ORF72-containing expanded G4C2 proteins.

Experimental data

-The authors present data showing that PR20 and GR20 added to an RRL in vitro system inhibits protein synthesis (Figure 1). These data are replotted to obtain an IC50 in panels c and d (I think?). Next the authors perform puromycin assays in E. coli (panel e). At this point, the authors have not demonstrated that these peptides inhibit translation in any bacterial system. These data are needed to provide a rationale for the use of E. coli as a model system for the puromycin assays. The authors cite evidence that DPRs interfere with mitochondrial translation as evidence that they use bacterial ribosome as a model system. The authors should perform studies using an in vitro system similar what the authors did in RRL in Figure 1 panels a-d?

-While the authors show that DPRs can bind in trans in vitro, they do not show that this is the mode of inhibition in cells. What is the evidence that the ribosome is NOT inhibited during translation of these peptides?

-(Minor) The data should be plotted in panel e of Figure 1 (bar graphs should not be used). Also why the space between 20 and 40?

-The peptidyl transfer reactions (Figure 1f and Supplementary Figure 1) should be replotted in order to observe shorter time points on the X axis because it appears the reaction is over after 5 mins? Further, the coloring in Figure 1f is confusing because in other panels GR is green but in this one panel, PR is green.

-The authors claim that poly-PR and poly-GR inhibit mammalian cell lysate as "strongly" as harringtonine and cycloheximide and then cite paper from the 1970s. Are the authors stating that both studies give similar IC50s? To my knowledge, there are also more relevant recent experiments that they should cite as well.

-The authors solve high-resolution cryoEM structures of PR20 and GR20 bound to yeast, rabbit and bacterial ribosomes. I'm not sure what the rationale is for solving the peptide bound to both yeast and rabbit. Did the authors think they should see species-specific differences? And at this point, there is no data that the peptides inhibit bacterial ribosomes.

-A few rRNA and ribosomal protein interactions are described to interact with the peptide in the exit tunnel but their significance is not discussed. How do other nascent chains in the peptide tunnels interact with the ribosome? Does PR20 and GR20 make different or similar interactions? It is described that PR20 stacks against 25S rRNA and uL4 "in keeping with strong binding of poly-PR". But to my knowledge, the thermodynamic behaviors of these peptides have not been addressed and therefore it is not clear that they do bind strongly. Additionally later in the text it is claimed "consistent with the high affinity of these DPR proteins in our biochemical experiments (Supplementary Fig. 2a,b,i)." But Supplementary Figure 2 is the cryo-EM flowchart, not binding experiments. Perhaps the authors meant Supplementary Figure 1 but again, no thermodynamic experiments are shown here to indicate that the peptides have high affinity.

-It also appears the interaction of PR20 with the rabbit and yeast ribosomes are different (compare panels b and e) but this is described as being similar in the text. This is confusing.

-In the structure of the rabbit 80S bound to the peptide (Figure 2e), there are two maps presented in the figure- one magenta and one purple. There is no way to tell the difference between these colors and the authors need to show this in a more transparent manner.

-While the authors show their map quality in Figure 2 (panels b-e), this inclusion means it is hard to see the interactions of the peptide with the ribosomal cavity. The colors don't help either. The map figures are appropriate in the supplementary data file but in the main text, they prevent a reader from understanding what the main points are of these interactions of the peptides with the exit tunnel.

-Related to this, how are the maps generated in all the structural figures presented in this manuscript?

-Next the authors describe structures of the peptides interacting with the bacterial ribosome (Figure 3). Again what evidence does the authors have that these peptides inhibit bacterial translation? This rationale is lacking and it seems these structures are tacked on to this manuscript. Also the manner in which the complex formation is explained is somewhat disingenuous. The authors form complexes by adding tRNA-fMet to the ribosome and load with a tRNA in the A site at the same time as the addition of the peptide (I think- please clarify). They then assert that that because of the high abundance of vacant 70S with the peptide that polyPR inhibitors prevent the formation of the initiation complex. This does not make sense because the authors did not perform initiation studies only simply added tRNA-

fMet to fill the P site of the 70S. All the cryoEM structure shows is that perhaps the peptide causes the dissociation of tRNAs. This is one of many statements that does not accurately explain their data which makes the mechanism of action of these peptides unclear.

-If the authors want to show that Ery relieves inhibition induced by PR20 (Figure 3d), the assays need to perform competition assays to a higher standard. Ery could simply bind more tightly to the ribosome than PR20 and that is why release is restored when both Ery and PR20 are present. But do these assays provide any insights into the mode of action of PR20 because they are performed with the E. coli ribosome? The authors state that these assays show that the peptide bind in the exit tunnel but what effect does PR20 have on bacterial translation?

- The authors cite a 1967 paper to describe how erythromycin affects the puromycin reaction. Has the Mankin group really not performed this assay using more modern approaches?

-The constriction of poly-PR in the tunnel (Figure 2) is explained as preventing A- and P-tRNA from binding PTC. But the authors do not show that this actually occurs and instead show that the peptidyl transferase activity is inhibited. The authors could biochemically test this hypothesis by looking at peptidyl-tRNA drop-off upon introduction of DPR. So it is unclear how these inhibitors work.

-The mechanism of these inhibitors is not clearly described because in some places, it is proposed that the peptides inhibits peptide bond formation, in the structures the peptides overlap with both A and P-site tRNAs indicating that they can't bind, and in other structures of just the large subunit without tRNA ligands, the peptides bind. This lack of clarity means it leaves the reader not knowing what is the endogenous role of these peptides. And though the authors state in the Abstract and elsewhere that these inhibitors have similar inhibitions as other antibiotics, it is not appropriate to compare them to such antibiotics unless there is evidence they function this way. Given the ambiguity of the conflicting data, it is not clear what they are doing. Likewise, another possibility is that they may inhibit during their own translation rather than functioning in trans. The authors could test this possibility.

-The authors further suggest that these peptides could be inhibiting translation during ribosome biogenesis and show overlays of biogenesis proteins bound at the nascent tunnel as evidence for this possibility (Supplementary Fig 4). But again, if they performed additionally assays (polysome profiles to look at subunit association defects), they could reconcile the peptides mode of action.

- As presented in the Discussion, many arginine-rich polypeptides do not cause translation inhibition. And after these structures, it is still not clear why these DPRs elicit inhibition. This is a really open question not addressed by these studies.

-The model figure is not warranted by the results presented here or otherwise known about this system. Many open questions remain that could be tested by the authors.

-(Minor) There is something wrong with the references

Reviewer 1

Overall, this work is presented as a very concise (even too much concise at parts) and clearly written manuscript. One of the strong sides of this work is the combination of a structural approach with the biochemical techniques, which only to some extent reveal the molecular mechanism of translation inhibition by poly-PR/GR peptides. In summary, the results of this work could, in principle, be interesting to a broad readership of *Nature Communication*. However, there are several major concerns denoted below that need to be addressed by the authors.

RESPONSE: We thank the reviewer for their positive comments and general support of our work. This manuscript was initially reviewed by *Nature* with a concise format. Upon the reviewer's comments here, we have rewritten the Introduction, Results and Discussion sections to describe the background and our findings in much greater detail. We have also expanded our biochemical and cryo-EM experiments, as described below, that provide additional support for ribosome inhibition by poly-PR and poly-GR.

1. One of the main conclusions of this work that poly-PR/GR peptides inhibit peptidyl transfer was made based on the results of puromycin assay on *E. coli* 70S and rabbit 80S ribosome. It is not surprising that binding of poly-PR/GR peptides to 70S/80S ribosomes decreases puromycin reactivity because these peptides simply block the access of puromycin to the A site of the ribosome. Therefore, puromycin assay, in this case, measures not the inhibition of peptidyl transfer reaction but rather the ability of puromycin to outcompete poly-PR/GR peptides from the ribosomal A site. In my opinion, this assay is inappropriate here because it does not allow us to discriminate between competition for the A site and the actual inhibition of PTC activity. Especially with the provided structures, it is pretty clear that these peptides prevent access of puromycin to the A site. Thus, it appears to be impossible to draw an unambiguous conclusion from the provided puromycin assay results.

RESPONSE: We agree with the reviewer that poly-PR and poly-GR most likely compete with puromycin or fMet-tRNA for binding in the PTC. We note that competition of poly-PR and poly-GR with the peptidyl-transfer substrate(s) is a novel finding.

The disagreement that the reviewer expresses perhaps reflects a difference in terms rather than mechanisms. Reactions may be inhibited by perturbation of substrate/transition-state positioning, or competition with substrate binding, the latter being the most common mechanism of inhibition of enzyme-catalyzed reactions. Our assays are consistent with the competitive inhibition of peptidyl transfer, as the reviewer notes.

To address the reviewer's concern and avoid potential confusion, we now state several times in the manuscript that our findings are consistent with *competitive* inhibition.

2. The models of translation inhibition by poly-PR/GR peptides are based almost exclusively on the structural data. However, a definitive answer about the molecular mechanism of inhibition could be provided only by functional studies. In my opinion, this type of data to support the claims should be provided in work published in such a high-rank scientific journal. For example, a simple toe-printing experiment that takes less than a day can reveal which stage of translation is being inhibited by the poly-PR/GR peptides and whether the ribosome initiation complexes can be assembled in the presence of these peptides or not.

RESPONSE: We thank the reviewer for this suggestion to examine the inhibition mechanism more deeply in the context of mRNA translation. To this end, we have performed an extensive set of experiments that we now describe in the manuscript.

To address the reviewer's question whether initiation or elongation are affected by poly-PR or poly-GR, we undertook sucrose-gradient polysome fractionation and kinetic assays in rabbit reticulocyte lysates and compared them to small-molecule inhibitors with known modes of action. These results are described in a new section of the manuscript and **Fig. 1** and **Supplementary Fig. 1**. The new findings suggest that poly-PR and poly-GR inhibit translation elongation while poly-PR can also inhibit at the initiation stage of translation.

These results are confirmed by the requested toeprinting experiments (**Supplementary Fig. 2**). We find that PR₂₀ is similar to the initiation inhibitor harringtonine in that it results in an accumulation of ribosomes near the start codon of beta-globin mRNA (n=2). GR₂₀ behaves more similarly to elongation inhibitors resulting in accumulation of ribosomes on the open reading frame.

We have rewritten the manuscript to include these novel results.

3. Based on some previously published data, the authors state that these peptides act in trans (a peptide is being synthesized on ribosome A but inhibits ribosome B), disfavoured the idea that they could also work in cis (inhibit and stall the very ribosome that synthesized them). Again, a simple toe-printing experiment can easily distinguish between these two possibilities. However, what I find even more important is that the suggested in trans mode of action of poly-PR/GR peptides is contradictory to their orientation in the ribosome. Most of the antibacterial PrAMP peptides that are very similar to the poly-PR/GR peptides studied here bind ribosomal tunnel in the reverse direction, with the C-terminus sticking into the tunnel. This explains how these PrAMPs could be synthesized by the ribosome, in the first place, and then how they inhibit other ribosomes in trans. Because poly-PR/GR peptides bind ribosome with their N-termini facing the tunnel, it is unclear how they could even be synthesized without the same ribosome being stalled due to their inhibitory in cis action in this case?

RESPONSE: We chose to investigate translation inhibition *in trans* as this mode has been shown in cellular studies and is correlated with global inhibition of cellular translation in cells. However, evidence also exists that poly-PR and poly-GR can inhibit translation *in cis*^{1,2}, as we discuss in the manuscript.

Most but not all PrAMPs bind in the C-terminus out direction. For example, Api137 has been shown by cryo-EM to bind with the N-terminus pointed out (same as a nascent chain)³. PrAMPs are translated as pro-peptides, which must be cleaved to become inhibitory. When we were inspecting our yeast 80S ribosome with PR₂₀ map, it was quite evident that the region of strongest density matched most closely to the PRP motif from Api137 with it is N-terminal out orientation. Since building away from this motif in this N-out conformation was easier and gave better peptide geometry and better agreement with the map, we modeled poly-PR in the N-out conformation.

To address the reviewer's concern, we emphasize evidence of both modes of inhibition in the Introduction and Discussion.

4. How sure are the authors that they actually observe the charge density maps corresponding to the peptides in the tunnel of the yeast 80S-GR₂₀ and rabbit 80S-PR₂₀ structures shown in figures 2d and 2e, respectively? Undoubtedly, there is a clear density for the peptides in the tunnel in the case of yeast 80S-PR₂₀ and E. coli 70S-PR₂₀ structures. However, the other two maps are questionable. For example, by closely looking at figure 2d, one can see that the putative density for the GR₂₀ peptide covers only a small portion of the peptide, and the model does not even fit it well. Even worse is the density for the peptide shown in Figure 2e, where only a short fragment of density for the peptide could be observed next to the PTC. Moreover, on Figures 2b, 2d, and 2e, the map for the peptide is depicted at 2.5sigma, whereas the map for the ribosomal parts is shown at 4sigma. This makes the density for the peptide look stronger than, in fact, it is – which is inappropriate. The authors should either remove the density for the ribosome and show only the density for the peptides or show the density for all parts at the same uniform contour level. Furthermore, in Figure 2e, there is an additional map shown at 1.1sigma for the middle portion of the peptide in the tunnel, which is clearly invisible at 2.5sigma. This definitely misleads because it makes an impression as if the density for the peptide is observed throughout the tunnel up to the constriction point, while there is only a short density fragment next to the PTC.

RESPONSE: We agree with the reviewer that the maps for the yeast 80S-PR₂₀ and the E. coli 70S-PR₂₀ complex were better than the yeast GR₂₀ map, despite our extensive efforts to resolve the heterogeneity by masked ML classification in Frealign and Relion. We hypothesize that heterogeneity may be due to GR₂₀ taking multiple conformations and/or orientations in the peptide tunnel. Nevertheless, the continuous features of the density suggest that it corresponds to GR₂₀ rather than other molecules or artefacts of data processing.

To further ameliorate these concerns, we have collected additional cryo-EM data (**Fig. 2, Supplementary Fig. 4**). We now report a structure for the rabbit 80S ribosome with GR₂₀, which features a slightly better density for poly-GR in the peptide tunnel. In addition, we collected data for a “negative control”, an identical preparation of the rabbit 80S ribosome without the addition of GR₂₀. The absence of density in the tunnel clearly indicates that density in the 80S*GR₂₀ data set belongs to GR₂₀.

We have added the new cryo-EM analyses to the manuscript. We also followed the reviewer’s suggestion to show maps at a uniform level in the figures. In figures with different contour levels, where different contour levels are labeled, we have also expanded the figure legends to state the use of different contour levels.

Relatively minor issues:

5. The authors used L-arginine as a negative control for their translation inhibition studies. A control should be an experiment with only a single parameter altered at a time. Therefore, the authors should have considered using poly-R, poly-G, and poly-P peptides of the same length.

RESPONSE: We ordered poly-R, poly-G and poly-P peptides of matching length from the same supplier as used for poly-PR and poly-GR and were told months later that the homo-peptides could not be made to the specified lengths. Also, it is not clear how to interpret the results from such an experiment. If poly-R, poly-G and/or poly-P were also found to inhibit translation, the result would be irrelevant to the ALS/FTD disease.

6. *The authors state that they checked the effect of poly-PR/GR peptides on puromycin reactivity with the 70S ribosomes from E.coli because these peptides were known to interfere with mitochondrial translation. I need to say that, maybe using the E.coli 70S ribosome as a model for mammalian mitochondrial 55S ribosomes would have been appropriate 30-40 years ago, it is not really the case anymore because these ribosomes are drastically different as demonstrated by numerous studies. Therefore, if the authors would like to make a claim about inhibition of mitochondrial ribosomes by the poly-PR/GR peptides, they should really use isolated mitochondrial ribosomes, not the bacterial ones.*

RESPONSE: We agree with the reviewer that bacterial and mitochondrial ribosomes might exhibit differences. We are not presently equipped to repeat these experiments with mitochondrial ribosomes. To address the reviewer's concern, we have reworded our discussion to tone down the parallels between the *E. coli* and mitochondrial ribosome.

7. *It is very confusing that the results of the puromycin assay with the 70S and 80S ribosomes shown in Figures 1e and 1f, respectively, are shown using completely different representations of basically the same assay. These two panels should have the same style of representation.*

RESPONSE: We have changed panel 1e and 1f (now **Fig. 2A and 2C**) to have the same style of representation.

8. *Although the authors claim that poly-PR/GR peptides are oriented with their N-termini towards the exit, there is no evidence in the manuscript to support this conclusion. Given that there are only short fragments of these longer peptide repeats visible in the tunnel, how is it even possible to determine such orientation? Is it possible that poly-PR/GR peptides bind ribosome in a reverse orientation similar to PrAMPs?*

RESPONSE: Please see our response to comment #3 regarding PrAMPs, some examples of which also bind in the N-out orientation. We now state in the Methods (In Section: Model building and refinement with subtitle: Yeast 80S model building) that the opposite direction is less compatible with the map.

9. *The experiment with erythromycin and E.coli 70S ribosomes was accomplished to show that poly-PR/GR peptides inhibit translation by binding at the specific site in the ribosomal tunnel and also to show that erythromycin and poly-PR/GR peptides have overlapping binding sites. But isn't this totally obvious from the structural studies presented here?*

RESPONSE: We agree with the reviewer that the competition experiment with erythromycin and poly-PR confirms the structural part of the study. It is possible, however, that strong inhibition demonstrated in biochemical experiments involved additional modes of poly-PR/GR action, such as non-specific binding to RNA, which could confer a large inhibitory effect. The competition experiment demonstrates the absence of other modes of potent inhibition. Using two lines of evidence therefore makes a stronger case that the functional inhibition of translation is directly due to binding of the DPR proteins in the tunnel.

10. *The figures showing the cryo-EM density maps should be made significantly bigger to allow the reader to examine the map without a magnifying glass.*

RESPONSE: Thank you. We have made the panels simpler in the main text and show EM density panels in Supplementary figures, where they can be further enlarged by zooming in (**Supplementary Fig. 3,4,5**).

11. *The authors provided almost no introduction at the beginning of the manuscript. Instead, they go straight to the “G4C2 repeats in the C9ORF72”. What are G4C2 repeats? What is C9ORF72? A few paragraphs spanning a page or two introducing this topic should be added to the manuscript in my opinion.*

RESPONSE: We thank the reviewer for the suggestion. We have written a new Introduction to describe the disease and the nature of the mutation in more detail.

Reviewer 2

1. *This manuscript presents biochemical and structural insights into how expanded G4C2 repeats located in C9ORF72 inhibit translation. The repeat length in C9ORF72 correlates to ALS/FTD pathologies and this manuscript aims to identify the molecular basis for this function. Overall this is an interesting manuscript but it is hard to assess the data given the shortness of the manuscript. Many important details of prior data that rationalizes their own work is also missing. It is also confusing how the authors use multiple different model systems without clear rationales for their experiments. Together this constitutes a significant weakness because the absence of such details coupled with a lack of rationale made it difficult to determine the impact of this work as discussed below.*

RESPONSE: We thank the reviewer for his/her overall positive assessment of our manuscript and also apologize for the lack of sufficient details and rationales in the original short format of the manuscript. In the revision, we have expanded the Introduction to describe the disease and the nature of the mutation in greater detail, outlining the rationale for this work. Moreover, we have revised the Results section substantially to include more rationales for each experiment, so that more readers could appreciate the relevance of our work.

2. *how do the repeats work? The authors perform all their assays with the impression that translated poly-PR and poly-GR can somehow rebind the ribosome after it has been translated and released. How it does this is unknown. The authors could perform biochemical assays to test this but do not. In other words, many cryo-EM structures are solved without a rationale and therefore the biological significance of some of these structures is minimal. Figures presented of the structures are not clear and one cannot discern the details the authors refer to in the text (Figure 2b-e; Figure 3b). Lastly, their model figure is not based upon the data presented or tested here and to my knowledge, not shown in other work. All of these issues are explained in more detail below.*

RESPONSE: Multiple lines of evidence from cell and tissue experiments show that these DPR proteins are translated and are distributed in cellular compartments. It has also been shown that they associate with ribosomes and inhibit global cellular translation. Because the mechanism of translation inhibition and binding was unknown, we undertook our biochemical (*in extracto* translation experiments and *in vitro* reconstituted peptidyl transfer assays in multiple systems) and structural approaches, which revealed how DPRs bind and inhibit ribosomes.

To address the reviewer's concern, we have revised our Introduction and discussion to more clearly describe the state of the art in the field. We have also carried out additional experiments and sought to clarify the figures, results and discussion to better articulate the mechanism.

3. *Minimal information-A number of the sections are incredibly short and lack details required to understand prior studies and the rationale and outcome of the experiments presented in this manuscript. For example, in the Introduction there are not enough details to understand previous experiments needed to know how to place these new experiments. For example, the authors describe experiments that demonstrate how poly-PR and poly-GR impair cellular translation but there are missing details. What system were these experiments performed in (animal model, ALS-FTD patients, cell culture etc)? Were these proteins overexpressed or do the authors mean that endogenous levels of these proteins/peptides alone affected translation? And finally what triggers the expression of these proteins in human patient brains? There is a sentence of how expression of poly-PR and poly-GR correlates with neurodegeneration but more information is needed to understand the precise correlation.*

RESPONSE: Thank you for this suggestion. We have rewritten the introduction to address the points raised by the reviewer.

4. *When the authors describe how “poly-PR” and “poly-GR” impair global cellular translation, do they mean C9ORF72 proteins containing an expansion of these repeats or did these assays simply rely on peptides containing ONLY contain PR or GR? (first sentence of the Results section- this information though should go into the Intro). What other parts of the protein are the authors removing in these assays? This is important because the authors in this manuscript only use PR or GR containing peptides but they have not set out the premise that it is acceptable to use these peptides as mimics of C9ORF72-containing expanded G4C2 proteins.*

RESPONSE: We apologize that the initial version did not clearly explain the relationship between the C9ORF72 gene and the translation-inhibiting DPR peptides. Briefly, GGGGCC repeat expansion occurs in the first intron of the C9ORF72 gene, so C9ORF72 protein is not altered by the disease mutation. Sense and antisense expanded GGGGCC repeats can be translated into dipeptide repeat (DPR) proteins, such as poly-GR and poly-PR. Numerous labs in the ALS/FTD field have studied the toxicity of these DPR peptides in animal models, cell culture and cell extracts. Here we undertake the in vitro and structural studies of DPR interactions with the ribosome.

To address this concern, we have rewritten the Introduction to better explain the disease and the nature of the mutation.

Experimental data

-The authors present data showing that PR20 and GR20 added to an RRL in vitro system inhibits protein synthesis (Figure 1). These data are replotted to obtain an IC50 in panels c and d (I think?). Next the authors perform puromycin assays in E.coli (panel e). At this point, the authors have not demonstrated that these peptides inhibit translation in any bacterial system. These data are needed to provide a rationale for the use of E. coli as a model system for the puromycin assays. The authors cite evidence that DPRs interfere with mitochondrial translation as evidence that they use bacterial ribosome as a model system. The authors should perform studies using an in vitro system similar what the authors did in RRL in Figure 1 panels a-d?

RESPONSE: We thank the reviewer for pointing out this logical leap. In the revised draft of the manuscript, we have now tested the DPR proteins using a commercial bacterial extract system (NEBExpress) (**Supplementary. Fig. 3A**).

-While the authors show that DPRs can bind in trans in vitro, they do not show that this is the mode of inhibition in cells. What is the evidence that the ribosome is NOT inhibited during translation of these peptides?

RESPONSE: In the revised introduction and discussion, we discuss the large body of evidence that DPRs are translated and released from ribosomes into other cellular and extracellular compartments (with relevant references). Moreover, transfection of DRPs or overexpression leads to similar observables, and we cite those studies as well. Global cellular inhibition of translation suggests that ribosome inhibition *in trans* is the predominant mode of action. Nevertheless, a recently published report shows that translation of DPRs is slowed or inhibited in a length dependent manner^{1,2}. We also cite that work and note that the *in trans* and *in cis* mechanisms of inhibition are not mutually exclusive.

-(Minor) The data should be plotted in panel e of Figure 1 (bar graphs should not be used). Also why the space between 20 and 40?

RESPONSE. Thank you for the suggestion. We have replotted the graph in old Fig. 1e (now **Fig. 2b**) as a scatter plot. The space between 20 and 40 occurs because we have DPR stocks of 4, 10, 20 or 40 repeats of PR and 4, 10, or 20 repeats of GR.

-The peptidyl transfer reactions (Figure 1f and Supplementary Figure 1) should be replotted in order to observe shorter time points on the X axis because it appears the react is over after 5 mins? Further, the coloring in Figure 1f is confusing because in other panels GR is green but in this one panel, PR is green.

RESPONSE. Thank you. We have fixed the colors of this figure, which is now **Fig. 2c**.

-The authors claim that poly-PR and poly-GR inhibit mammalian cell lysate as “strongly” as harringtonine and cycloheximide and then cite paper from the 1970s. Are the authors stating that both studies give similar IC50s? To my knowledge, there are also more relevant recent experiments that they should cite as well.

RESPONSE: Despite a deep literature dive, we failed to find more recent papers measuring IC50 in cell lysates. More recent work tends to focus on cells and cellular IC50 which may not translate to lysates. Thus, we measured the IC50 of harringtonine and cycloheximide in the same extracts used to perform the DPR assay. These reveal that the IC50 of cycloheximide and harringtonine are in the range of 0.01-0.5 μ M, and we have added these data for **Fig. 1**.

-The authors solve high-resolution cryo-EM structures of PR20 and GR20 bound to yeast, rabbit and bacterial ribosomes. I’m not sure what the rationale is for solving the peptide bound to both yeast and rabbit. Did the authors think they should see species-specific differences? And at this point, there is no data that the peptides inhibit bacterial ribosomes.

RESPONSE: We started with yeast ribosome because we had the tools to determine high resolution structure and used it to determine the feasibility of visualizing these DPRs bound at a specific site. As these yeast structures demonstrated poly-PR and poly-GR binding to the

peptide tunnel, we developed the capability to assemble the more relevant, mammalian (rabbit) ribosome complexes and had included one such structure with PR₂₀ in the first version of the manuscript. Unfortunately, the resolution of the rabbit ribosome structures is not as high as the yeast, and so both were included.

In the revised manuscript we include 2 additional rabbit ribosome datasets: GR₂₀ and a buffer only control. These data show occupancy of the rabbit ribosome tunnel by GR₂₀ and strongly support our original observations with the yeast ribosome. In addition, because much the biochemistry was carried out with the bacterial ribosome, we performed a peptidyl transfer reaction catalyzed by elongation factors and amino-acyl tRNAs *in vitro* and imaged it by cryo-EM in the presence or absence of PR₂₀. In the original manuscript we did not discuss the control experiment in which we repeated the reaction without PR₂₀ and this is now included (**Supplementary Fig. 6**). Analysis of the distribution of ribosomal states in the presence or absence PR₂₀ suggests that PR₂₀ perturbs peptide bond formation when bound to the tunnel. Thus, our data strongly support translation inhibition by DPRs is mediated by binding to the peptide tunnels of ribosomes from different species.

-A few rRNA and ribosomal protein interactions are described to interact with the peptide in the exit tunnel but their significance is not discussed. How do other nascent chains in the peptide tunnels interact with the ribosome? Does PR20 and GR20 make different or similar interactions?

RESPONSE: The description of PR₂₀ and GR₂₀ interaction with these residues is meant to help the reader orient their position in the tunnel. PR₂₀ and GR₂₀ are similar in that they interact with peptide tunnel constrictions via stacking interactions with bases and polar interaction with phosphate backbone, as expected, however the details of interactions are different due to the rigidity of proline residues in PR₂₀ and high dynamics of glycine residues in GR₂₀. It is also interesting to compare PR₂₀ and GR₂₀ to known translation inhibitors like the PrAMPs (e.g. Api137) as we discuss in the Discussion section. Parts of PR₂₀ make very similar interactions to Api137 as shown in **Supplementary Fig. 7a-b**.

It is described that PR20 stacks against 25S rRNA and uL4 “in keeping with strong binding of poly-PR”. But to my knowledge, the thermodynamic behaviors of these peptides have not been addressed and therefore it is not clear that they do bind strongly. Additionally later in the text it is claimed “consistent with the high affinity of these DPR proteins in our biochemical experiments (Supplementary Fig. 2a,b,i).” But Supplementary Figure 2 is the cryo-EM flowchart, not binding experiments. Perhaps the authors meant Supplementary Figure 1 but again, no thermodynamic experiments are shown here to indicate that the peptides have high affinity.

RESPONSE: We apologize for the confusion this sentence raised. The callout to Sup. Fig. 2a,b,i was referring to the earlier clause of the sentence relating to cryo-EM. The high affinity of PR₂₀ and GR₂₀ is implied by high IC₅₀ for translation and peptidyl transfer inhibition and we now state this in the text. We would also like the reviewer to know that we sought to directly measure PR₂₀ and GR₂₀ binding affinities to the ribosome by various approaches but Western-blotting for the DPRs were made impossible by the cross-reactivity of ribosomal proteins (which are also very positively charged) with anti-poly-PR and anti-poly-GR antibodies.

-It also appears the interaction of PR20 with the rabbit and yeast ribosomes are different (compare panels b and e) but this is described as being similar in the text. This is confusing.

RESPONSE: The interactions of PR₂₀ with the rabbit and yeast ribosome are most similar at the constrictions of the peptide tunnel. The visual differences in the figure were due to the lower resolution of the rabbit ribosome map due to the different instruments used to collect the datasets. To quantify the similarity between the rabbit and yeast ribosome PR₂₀ interactions, we have measured the RMSD of the PR₂₀ backbone between the two structures and find that it is 1.67 Å.

-In the structure of the rabbit 80S bound to the peptide (Figure 2e), there are two maps presented in the figure- one magenta and one purple. There is no way to tell the difference between these colors and the authors need to show this in a more transparent manner.

RESPONSE: Thank you for this suggestion. Our attempts to show the data as clearly as possible have led to confusion. To rectify this situation, we have moved the density figures from the main text and show additional density views in **Supplementary Fig. 5**.

-While the authors show their map quality in Figure 2 (panels b-e), this inclusion means it is hard to see the interactions of the peptide with the ribosomal cavity. The colors don't help either. The map figures are appropriate in the supplementary data file but in the main text, they prevent a reader from understanding what the main points are of these interactions of the peptides with the exit tunnel.

RESPONSE: Thank you. We now show cleaner figures of the structures without density, and we show density views in the supplement as discussed above.

-Related to this, how are the maps generated in all the structural figures presented this manuscript?

RESPONSE: As described in Methods, the maps were generated by cryo-EM data processing with the widely used software package FREALIGN. The figures are generated in Pymol as stated in the methods. We have expanded the figure legends for density view figures to make more clear which map is shown and what settings were used (for example what B-factor has been applied to the map, whether low-pass filtering was used, and what sigma level the map is shown at).

-Next the authors describe structures of the peptides interacting with the bacterial ribosome (Figure 3). Again what evidence does the authors have that these peptides inhibit bacterial translation? This rationale is lacking and it seems these structures are tacked on to this manuscript.

RESPONSE: We agree that there is no evidence that these DPRs occur in bacteria. However, we used bacterial ribosomes as the best initial model system for biochemical studies, as they remain a useful and relevant tool for studying translation due to the ribosome's conservation. Furthermore, we used bacterial ribosomes as a tool to understand whether the peptide tunnel binding site that we observe in all the different cryo-EM structures is relevant to the translation inhibition measured *in vitro* and previously reported in cells. To this end, we used the well-known bacterial translation inhibitor erythromycin that binds the bacterial peptide tunnel, to compete with the PR₂₀ for peptide tunnel binding (see **Fig. 4** and newly expanded discussion). We find that erythromycin does compete with PR₂₀ for peptide tunnel binding thus substantiating the relevance of the DPRs binding sites we identify using cryo-EM.

Also the manner in which the complex formation is explained is somewhat disingenuous. The authors form complexes by adding tRNA-fMet to the ribosome and load with a tRNA in the A site at the same time as the addition of the peptide (I think- please clarify). They then assert that that because of the high abundance of vacant 70S with the peptide that polyPR inhibitors prevent the formation of the initiation complex. This does not make sense because the authors did not perform initiation studies only simply added tRNA-fMet to fill the P site of the 70S. All the cryoEM structure shows is that perhaps the peptide causes the dissociation of tRNAs. This is one of many statements that does not accurately explain their data which makes the mechanism of action of these peptides unclear.

RESPONSE: Thank you for pointing this out. The procedure for preparing the 70S ribosome complexes is described in detail in Methods. Briefly, 30S and 50S subunits were mixed with mRNA for 10 minutes. Next fMet-tRNA^{fMet} was added and after 3 minutes PR₂₀ was added for 5 minutes. The reaction was then held on ice until the cryo-plunging apparatus was prepared. Next, the ribosome complex and a ternary complex of Val-tRNA^{Val}•EF-Tu•GTP were mixed and incubated for 10 minutes to allow the elongation step to complete prior to cryo-plunging. In the control dataset, there are less ribosomes without A site tRNA (20% instead of 34%).

Classification of the datasets shows that in the presence of poly-PR fewer ribosomes proceed through the peptidyl transfer reaction. Those that do, do not have strong poly-PR density in the tunnel. Ribosomes that are stalled have strong poly-PR density (**Supplementary Fig. 6**).

We have updated the discussion to focus on the peptidyl transfer inhibition, which the presented cryo-EM reaction represents, and toned down the suggestion that initiation could be affected.

-If the authors want to show that Ery relieves inhibition induced by PR20 (Figure 3d), the assays need to perform competition assays to a higher standard. Ery could simply bind more tightly to the ribosome than PR20 and that is why release is restored when both Ery and PR20 are present. But do these assays provide any insights into the mode of action of PR20 because they are performed with the E. coli ribosome? The authors state that these assays show that the peptide bind in the exit tunnel but what effect does PR20 have on bacterial translation?

RESPONSE: We note that the bacterial system was only used as a model system for the erythromycin competition experiment. Erythromycin and other macrolides are antimicrobial inhibitors, and they are not suitable for a similar experiment with eukaryotic ribosomes because they do not bind eukaryotic ribosomes. We have added data to show that PR₂₀ inhibits bacterial translation in an extract system (see **Supplementary Fig. 3A**).

As we state in the rationale for this experiment, its goal is to demonstrate that binding of the DPRs in the tunnel is the primary mode of inhibition, as opposed to a strong non-specific inhibition such as binding to mRNA or tRNA. Thus, Ery binding to the peptide tunnel preventing PR₂₀ binding is exactly the feature that the take the advantage of to demonstrate that PR₂₀ binding in the tunnel is responsible for ribosome inhibition.

To address this concern, we have expanded the section of the manuscript to make the logic of this experiment clearer.

- The authors cite a 1967 paper to describe how erythromycin affects the puromycin reaction. Has the Mankin group really not performed this assay using more modern approaches?

RESPONSE: The 1967 paper that we cite is the earliest, to our knowledge, demonstration of the independence of puromycin reaction on erythromycin. There are more recent papers that use erythromycin and puromycin, taking advantage of the same mode of action that was demonstrated in 1967. We have now added a reference to one of the most relevant papers, but unfortunately, we could not cite many other articles using a similar assay due to the journal's limiting the number of references.

-The constriction of poly-PR in the tunnel (Figure 2) is explained as preventing A- and P-tRNA from binding PTC. But the authors do not show that this actually occurs and instead show that the peptidyl transferase activity is inhibited. The authors could biochemically test this hypothesis by looking at peptidyl-tRNA drop-off upon introduction of DPR. So it is unclear how these inhibitors work.

RESPONSE: Thank you. We now clarify that the binding of poly-PR and poly-GR to the peptide tunnel interferes with A and P-tRNA binding, as evidenced by several observations.

1) We have solved an additional structure of the rabbit 80S bound with GR₂₀, and a structure of rabbit 80S in the absence of any DPR (negative control). The percentage of ribosomes in the rotated state with P/E tRNA held out of the PTC increases 2-fold for with PR₂₀ and 1.5-fold with GR₂₀ (first row of table below). Additionally, non-rotated ribosomes with GR₂₀ and strong density for P tRNA ASL on the 40S exhibit very weak density for the acceptor arm of P-tRNA in the 60S subunit indicating that the P-tRNA is dynamic in the presence of the DPRs and is not stably bound in the PTC (22% (see 3rd row below).

	Buffer	PolyPR	PolyGR
Rotated, P/E-tRNA	12.3	25.0	18.9
Non-rotated, P/P tRNA	80.9	66.4	52.1
Non-rotated, weak P/P tRNA	0.0	0.0	22.1
60S	6.8	4.9	6.9
40S	0.0	3.7	0.0

2) Similarly, the cryo-EM reaction of bacterial 70S with and without poly-PR reveals that in the presence of poly-PR, fewer ribosomes proceed through the peptidyl transfer reaction. Those that do, do not have strong poly-PR density in the tunnel. Ribosomes that are stalled have strong poly-PR density (see **Supplementary Fig. 6**).

3) We further investigated the mechanism of inhibition using sucrose gradient assays that reveal that poly-PR and poly-GR treatment of translation reactions leads to the accumulations of halfmers on mRNA (**Fig. 1 and Supplementary Fig. 1**). Halfmers occur when 40S subunits that are bound to mRNA are delayed in joining a 60S subunit.

4) We now include extensive data using polyribosome profiles and toeprinting, indicating inhibition of initiation and elongation (see response to comment #2 of Reviewer 1).

-The mechanism of these inhibitors is not clearly described because in some places, it is

proposed that the peptides inhibits peptide bond formation, in the structures the peptides overlap with both A and P-site tRNAs indicating that they can't bind, and in other structures of just the large subunit without tRNA ligands, the peptides bind. This lack of clarity means it leaves the reader not knowing what is the endogenous role of these peptides. And though the authors state in the Abstract and elsewhere that these inhibitors have similar inhibitions as other antibiotics, it is not appropriate to compare them to such antibiotics unless there is evidence they function this way. Given the ambiguity of the conflicting data, it is not clear what they are doing. Likewise, another possibility is that they may inhibit during their own translation rather than functioning in trans. The authors could test this possibility.

RESPONSE: We apologize that this confusion may have arisen from our brief introduction in the initial version of the paper. We have now expanded the introduction to clarify that the DPRs have no endogenous role but are instead translated by a disease-related process. In the revised introduction we discuss the wide body of evidence that shows that DPRs are translated and released from ribosomes into other cellular and extracellular compartments. Moreover, transfection of DPRs and overexpression of DPRs leads to similar observables. Thus, we chose to study ribosome inhibition in trans. There is a published report ¹ that shows that translation of DPRs is slowed or inhibited in a length dependent manner, which we cite. The *in trans* and *in cis* mechanisms are not mutually exclusive.

Next we seek to address the second point that it is unclear what the peptides are doing. We have added new experiments in extracts to compare the mode of this inhibition to known translation initiation, elongation and termination inhibitors in the expanded manuscript. Our peptidyl transfer experiments show that this translation inhibition is due to problems at the peptidyl transferase center rather than another part of the ribosome. The cryo-EM experiments show that the problems at the peptidyl transferase center may be due to steric clashes between the DPRs and the tRNA substrates at the peptidyl transferase center. These observations are in perfect agreement with each other and not in conflict. The fact that the same binding site may cause problems during translation initiation and elongation is also not unprecedented (cycloheximide can stall translation at the first and all subsequent steps of elongation). To substantiate that multiple translation processes can be affected by poly-PR and poly-GR and identify which processes they are, we greatly our biochemistry section to include extra controls polysome profiling, and toeprinting. (Fig. 1 and Supplementary Fig. 1, 2).

-The authors further suggest that these peptides could be inhibiting translation during ribosome biogenesis and show overlays of biogenesis proteins bound at the nascent tunnel as evidence for this possibility (Supplementary Fig 4). But again, if they performed additionally assays (polysome profiles to look at subunit association defects), they could reconcile the peptides mode of action.

RESPONSE: We speculated that ribosome biogenesis may be affected by these DPRs. Indeed, previous work has shown that the DPRs localize to the site of ribosome biogenesis in cells, the nucleolus ⁴⁻⁶. Previous work has also shown that ribosome biogenesis is affected by carrying out the experiment the reviewer proposes ⁷. A deeper understanding of how DPRs impact biogenesis however would require a thorough study that is not the focus of this work. In the revised manuscript we have moved mentioning of ribosome biogenesis to the Supplementary Information and limit it to drawing parallels with previously published work.

- As presented in the Discussion, many arginine-rich polypeptides do not cause translation inhibition. And after these structures, it is still not clear why these DPRs elicit inhibition. This is a really open question not addressed by these studies.

RESPONSE: We apologize that comparison of the DPRs with some previously tested arginine-rich polypeptides was not clearly explained. Experiments by us and previous studies by other labs show that translation is inhibited by poly-PR and poly-GR. Kanekura et al. previously also showed that PR₂₀ inhibits translation but that some cell-penetrating peptides that are rich in arginines such as TAT (GRKKRRQRRRPPQ), R12 (RRRRRRRRRRRR) and FHV (RRRRNRTRRRRRRVR) do not inhibit translation even at 100 μM in concentration⁸. We speculate that the different protein sequences particularly their shorter lengths and closer spacing of Arginine residues preclude their efficient entry into or binding of the ribosome's peptide exit tunnel. Our cryo-EM structures and the erythromycin experiments show that translation inhibition by poly-PR is dependent on binding to the peptide tunnel. We have reworked our introduction and discussion to make this conclusion clearer.

-The model figure is not warranted by the results presented here or otherwise known about this system. Many open questions remain that could be tested by the authors.

RESPONSE: We agree that there are open questions that will be tested by future studies. Our previous and new data point at the translation steps that are inhibited, which we present in the model figure updated in the light of our expanded results.

-(Minor) There is something wrong with the references.

RESPONSE: Thank you. We have corrected the references.

- 1 Radwan, M. *et al.* Arginine in C9ORF72 dipolypeptides mediates promiscuous proteome binding and multiple modes of toxicity. *Molecular & Cellular Proteomics*, doi:10.1074/mcp.RA119.001888 (2020).
- 2 Park, J. *et al.* ZNF598 co-translationally titrates poly(GR) protein implicated in the pathogenesis of C9ORF72-associated ALS/FTD. *Nucleic Acids Research*, gkab834, doi:10.1093/nar/gkab834 (2021).
- 3 Florin, T. *et al.* An antimicrobial peptide that inhibits translation by trapping release factors on the ribosome. *Nat Struct Mol Biol* **24**, 752-757, doi:10.1038/nsmb.3439 (2017).
- 4 Kanekura, K. *et al.* Characterization of membrane penetration and cytotoxicity of C9orf72-encoding arginine-rich dipeptides. *Sci Rep* **8**, 12740, doi:10.1038/s41598-018-31096-z (2018).
- 5 Lee, K. H. *et al.* C9orf72 Dipeptide Repeats Impair the Assembly, Dynamics, and Function of Membrane-Less Organelles. *Cell* **167**, 774-788 e717, doi:10.1016/j.cell.2016.10.002 (2016).
- 6 Tao, Z. *et al.* Nucleolar stress and impaired stress granule formation contribute to C9orf72 RAN translation-induced cytotoxicity. *Human Molecular Genetics* **24**, 2426-2441, doi:10.1093/hmg/ddv005 (2015).

- 7 Kwon, I. *et al.* Poly-dipeptides encoded by the C9orf72 repeats bind nucleoli, impede RNA biogenesis, and kill cells. *Science* **345**, 1139-1145, doi:10.1126/science.1254917 (2014).
- 8 Kanekura, K. *et al.* Poly-dipeptides encoded by the C9ORF72 repeats block global protein translation. *Hum Mol Genet* **25**, 1803-1813, doi:10.1093/hmg/ddw052 (2016).

Reviewers' Comments:

Reviewer #1:

Remarks to the Author:

After carefully reading the revised version of the manuscript, I was glad to see that the authors did make the manuscript much better! Especially with the new biochemical experiments, the work started to look like a much more coherent story. Most of the critical points raised by this reviewer were satisfactorily addressed by the authors in the revised version. However, there are still a few minor critical points/comments:

1. It is unclear from the introduction how the expansion of the GGGGCC repeats in the intron of the C9ORF72 ORF can result in the appearance of poly-PR and/or poly-GR tracts in the translated protein, given that introns are normally untranslated by definition.
2. There are no units of measure in the plots shown in Figure 1C.
3. Please add cryo-EM maps for the three structures of the DPR peptides shown in Figures 3B, D, E, so that it will be clear to the reader how well the models fit the observed cryo-EM map.
4. Since the authors are "selling" poly-PR/GR peptides as inhibitors of eukaryotic translation and mammalian translation, in particular, any experiments using bacterial ribosomes appear to be irrelevant and/or disconnected from the main story. Therefore, I would like to suggest to the authors to move Figure 4 to the Supplementary, as it does not reveal anything conceptually new compared to Figure 3.
5. Supplementary Figure 2, showing results of the toe-printing analysis, sheds new light on the mechanism of inhibition of ribosome by poly-PR/GR peptides and, in my opinion, deserves a spot in the main text.

We very much appreciate the positive review and the minor criticisms that have helped us to further improve the manuscript.

Reviewer's criticisms:

After carefully reading the revised version of the manuscript, I was glad to see that the authors did make the manuscript much better! Especially with the new biochemical experiments, the work started to look like a much more coherent story. Most of the critical points raised by this reviewer were satisfactorily addressed by the authors in the revised version. However, there are still a few minor critical points/comments:

RESPONSE: We thank the reviewer that he/she finds the manuscript much improved. We address the minor points below.

1. It is unclear from the introduction how the expansion of the GGGGCC repeats in the intron of the C9ORF72 ORF can result in the appearance of poly-PR and/or poly-GR tracts in the translated protein, given that introns are normally untranslated by definition.

RESPONSE: Thank you for pointing out that this was unclear in our introduction. We now have expanded the introduction by adding a sentence with three references to research that clarifies that the GGGGCC repeats-containing RNA is found in the cytoplasm and is associated with polysomes.

2. There are no units of measure in the plots shown in Figure 1C.

RESPONSE: Thank you for catching this error. Corrected.

3. Please add cryo-EM maps for the three structures of the DPR peptides shown in Figures 3B, D, E, so that it will be clear to the reader how well the models fit the observed cryo-EM map.

RESPONSE: We did include the maps in the original figure 3 of our initial submission, however, during the first round of review, one of the reviewers asked us to move the maps to Supplementary Figures for the clarity of main-text figures. We now added a statement in the

Legends for Figure 3B, D, and E (now Fig. 4B, D and E) that the models' fit to the cryo-EM maps are shown in several panels in Supplementary Figures 3 and 5.

4. Since the authors are “selling” poly-PR/GR peptides as inhibitors of eukaryotic translation and mammalian translation, in particular, any experiments using bacterial ribosomes appear to be irrelevant and/or disconnected from the main story. Therefore, I would like to suggest to the authors to move Figure 4 to the Supplementary, as it does not reveal anything conceptually new compared to Figure 3.

RESPONSE: We agree that demonstration of bacterial translation inhibition by poly-PR/GR is of secondary interest on its own. The main message of Figure 4, however, is that the inhibitory effect of poly-PR/GR can be outcompeted by erythromycin. Erythromycin is a bacterial inhibitor, so this experiment can only be done on the bacterial 70S ribosome. This experiment provides direct evidence that peptidyl-transfer inhibition is due to poly-PR/GR binding to the peptide tunnel. We therefore believe that this important result should be shown in a main figure. To further clarify the importance of these data, we have changed the title of this section to “Cryo-EM and competition assay of PR₂₀ on 70S ribosomes corroborate direct PTC inhibition”

5. Supplementary Figure 2, showing results of the toe-printing analysis, sheds new light on the mechanism of inhibition of ribosome by poly-PR/GR peptides and, in my opinion, deserves a spot in the main text.

RESPONSE: We thank the reviewer for appreciating the importance of new our toe-printing results. We now show them in main-text Figure 2.